# Principled Data Selection for Alignment: The Hidden Risks of Difficult Examples

Chengqian Gao[†1]   Haonan Li[1]   Liu Liu[2]   Zeke Xie[3]   Peilin Zhao[2 4]   Zhiqiang Xu[1]

## Abstract

The alignment of large language models (LLMs) often assumes that using more clean data yields better outcomes, overlooking the match between model capacity and example difficulty. Challenging this, we propose a new principle: *"Preference data vary in difficulty, and overly difficult examples hinder alignment, by exceeding the model's capacity."* Through systematic experimentation, we validate this principle with three key findings: (1) preference examples vary in difficulty, as evidenced by consistent learning orders across alignment runs; (2) overly difficult examples significantly degrade performance across four LLMs and two datasets; and (3) the capacity of a model dictates its threshold for handling difficult examples, underscoring a critical relationship between data selection and model capacity. Building on this principle, we introduce *Selective DPO*, which filters out overly difficult examples. This simple adjustment improves alignment performance by 9-16% in win rates on the AlpacaEval 2 benchmark compared to the DPO baseline, surpassing a series of DPO variants with different algorithmic adjustments. These results together illuminate the importance of aligning data difficulty with model capacity, offering a transformative perspective for improving alignment strategies in LLMs. Code is available at `https://github.com/glorg ao/SelectiveDPO`

## 1. Introduction

Data selection focuses on identifying the most valuable subset from a dataset while excluding ineffective samples (Albalak et al., 2024). It significantly improves the first two

[†]This work is done when Chengqian Gao works as an intern in Tencent AI Lab. [1]MBZUAI   [2]Tencent Inc   [3]HKUST (Guangzhou) [4]SJTU. Correspondence to: Liu Liu <leonliuliu@tencent.com>, Peilin Zhao <masonzhao@tencent.com>, Zhiqiang Xu <zhiqiang.xu@mbzuai.ac.ae>.

*Proceedings of the 42nd International Conference on Machine Learning*, Vancouver, Canada. PMLR 267, 2025. Copyright 2025 by the author(s).

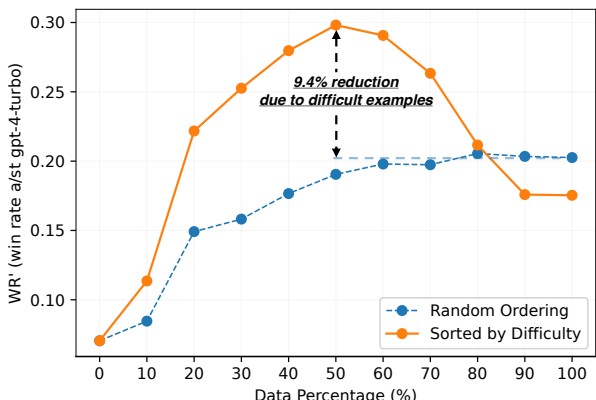

Figure 1: **Overly difficult examples hinder the alignment.** Training on difficult examples, identified by high *validation loss*, adversely affects alignment and decreases overall performance by 9.4% in win rate. The results are from experiments with four SFT models on the *UltraFeedback-binarized* dataset, *i.e.,* Figure 3.

stages of training large language models (LLMs): pretraining (Lee et al., 2021; Penedo et al., 2023; Tang et al., 2024) and supervised fine-tuning (SFT) (Cao et al., 2023; Qin et al., 2025; Zhou et al., 2023), by adhering to well-established principles. However, in the third stage, *i.e.,* preference alignment (Askell et al., 2021; Weidinger et al., 2021), data selection principles are often *implicit and superficial*, potentially limiting the alignment between LLM outputs and human preferences.

Prior studies in alignment underscore the importance of selecting error-free data by demonstrating the presence and negative impacts of mislabeled data (Wang et al., 2024a; Gao et al., 2024), noisy feedback (Mitchell, 2023; Chowdhury et al., 2024), and data with low agreement (Argilla, 2024), implicitly assuming that all error-free data are beneficial regardless of the model's capacity. However, we argue this assumption overlooks the relationship between data difficulty and model capacity. Our experiments show that overly difficult examples not only fail to improve alignment but can actually hinder the performance (see Figure 1). This observation motivates our systematic investigation into how example difficulty affects alignment performance.

Our main contribution is a new principle for preference data selection, which emphasizes the match between model

capacity and example difficulty:

> *Preference data vary in difficulty, and overly difficult examples hinder alignment, by exceeding the model's capacity.*

This principle has three key claims: (1) preference data can be categorized by difficulty levels, (2) overly difficult examples can harm alignment performance, and (3) difficulty is relative to the model's capacity—larger models, with greater capacity, can benefit from more difficult examples. We validate this principle through systematic experiments. Specifically:

**Preference examples vary in difficulty level (Section 3).** We show that, in DPO (Rafailov et al., 2024), the order in which testing examples are correctly classified by the implicit reward model is consistent across different runs and training data. This robust ordering reflects the existence of inherent example difficulties. Based on this observation, we use *validation loss* as a computational proxy to systematically identify and rank example difficulty.

**Difficult examples hinder alignment (Section 4).** We reveal that difficult examples–identified by high validation loss–significantly hinder alignment. Our experiments across two datasets and four pre-trained models show consistent performance drops when including these difficult examples. These challenging examples emerge naturally during data collection, rather than through artificial construction. This highlights the imperfections of the previous principle and calls for a new data selection principle for alignment tasks.

**Difficult examples exceed the model's capacity (Section 4).** We demonstrate that example difficulty interacts directly with model capacity. Experiments with models of 3B, 8B, and 14B parameters show that larger models benefit from higher proportions of difficult examples, confirming that difficulty must be calibrated to the model's capacity.

**Filtering out overly difficult examples yields remarkable gains (Section 5 and 6).** Finally, we validate our principle with a new method, *Selective DPO*, which filters out overly difficult examples. This approach achieves a 9–16% higher win rate on AlpacaEval 2 (Dubois et al., 2024) compared to standard DPO (Rafailov et al., 2024), outperforming state-of-the-art methods such as SimPO (Meng et al., 2024) and R-DPO (Park et al., 2024) while maintaining better perplexity and implicit reward margins.

## 2. Preliminaries

### 2.1. Preference Alignment with DPO

Preference alignment (Ouyang et al., 2022) aims to align the outputs of LLMs with human ethics and styles, ensuring that these models are safe, reliable, and effective for real-world applications (Christiano et al., 2017). In this study,

we focus on direct preference optimization (DPO) (Rafailov et al., 2024), a method known for its simplicity and robust performance in alignment tasks (Dubey et al., 2024). DPO trains a policy model, $\pi_{\boldsymbol{\theta}}$, on a dataset $\mathcal{D}$ containing prompt $x$, preferred response $y_w$, and rejected response $y_l$. The training objective incorporates a *reference SFT model*, $\pi_{\text{ref}}$, and a hyperparameter, $\beta$, to control the divergence between $\pi_{\boldsymbol{\theta}}$ and $\pi_{\text{ref}}$:

$$\mathcal{L}_{\text{DPO}}(\pi_{\boldsymbol{\theta}}, \mathcal{D}) = -\mathbb{E}_{(x,y_w,y_l)\sim\mathcal{D}}\Big[ \tag{1}$$
$$\log\sigma\big(\beta\log\frac{\pi_{\boldsymbol{\theta}}(y_w|x)}{\pi_{\text{ref}}(y_w|x)} - \beta\log\frac{\pi_{\boldsymbol{\theta}}(y_l|x)}{\pi_{\text{ref}}(y_l|x)}\big)\Big].$$

### 2.2. Quantifying the Example Difficulty

**Learned step as a measure of difficulty.** An example's *learned step* is defined as the earliest training step after which the model reliably distinguishes preferred responses from rejected answers. This is formalized as:

$$\text{LS}(x, y_w, y_l) = \min_{t_{\text{lrn}}}\Big\{ \tag{2}$$
$$t_{\text{lrn}}\Big|\beta\log\frac{\pi_{\boldsymbol{\theta}_t}(y_w|x)}{\pi_{\text{ref}}(y_w|x)} - \beta\log\frac{\pi_{\boldsymbol{\theta}_t}(y_l|x)}{\pi_{\text{ref}}(y_l|x)} > \delta, \forall t > t_{\text{lrn}}\Big\}.$$

A similar metric has been explored by Wu et al. (2021). The difference is that we calculate Eq. (2) exclusively on *held-out examples*, ensuring it reflects intrinsic difficulty rather than the order of data presentation (Zhu et al., 2024a). Larger learned steps indicate more difficult examples. For all experiments, we set $\delta = 0.4$.

**Validation loss as an alternative difficulty proxy.** We borrow *validation loss* (Wu et al., 2021; Rampp et al., 2024) as a computationally cheaper alternative to the *learned step*. Specifically, for a specific example $(x, y_w, y_l)$ from $\mathcal{D}\backslash\hat{\mathcal{D}}$, validation loss is defined as:

$$\text{VL}(x, y_w, y_l) = \tag{3}$$
$$-\log\sigma\big(\beta\log\frac{\pi_{\hat{\boldsymbol{\theta}}}(y_w|x)}{\pi_{\text{ref}}(y_w|x)} - \beta\log\frac{\pi_{\hat{\boldsymbol{\theta}}}(y_l|x)}{\pi_{\text{ref}}(y_l|x)}\big),$$

where $\pi_{\hat{\boldsymbol{\theta}}} = \arg\min_{\pi_{\boldsymbol{\theta}}} \mathcal{L}_{\text{DPO}}(\pi_{\boldsymbol{\theta}}, \hat{\mathcal{D}})$ is a *reference model*[1] trained on the subset $\hat{\mathcal{D}} \subset \mathcal{D}$. Low validation losses indicate easier examples. To compute the validation loss, we partition $\mathcal{D}$ equally into $\hat{\mathcal{D}}$ and $\mathcal{D} \setminus \hat{\mathcal{D}}$, train on one partition, evaluate on the other, and finally output average results over three runs.

---

[1] Throughout this work, the term reference model ($\pi_{\hat{\boldsymbol{\theta}}}$) refers to the aligned model trained on subsets of the training examples, while reference SFT model ($\pi_{\text{SFT}}$) specifically denotes the model defined in Eq. (1).

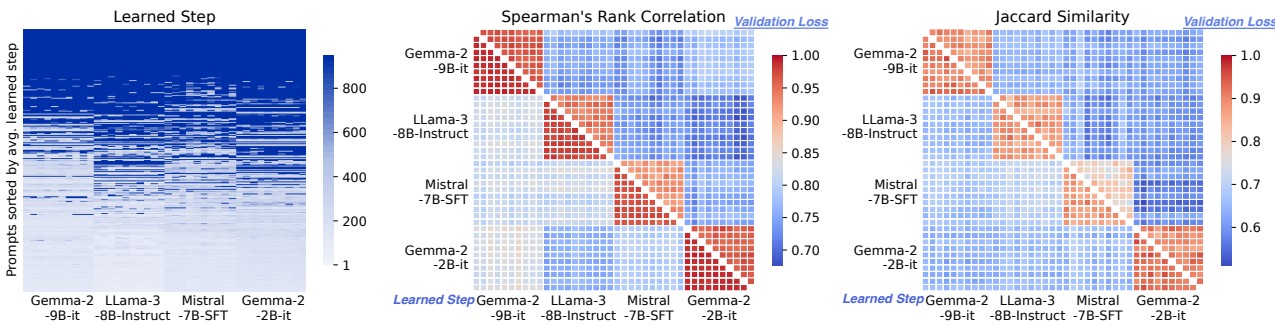

Figure 2: **Examples are learned in consistent orders across different runs of the same LLM**, despite variations in the training data and random seeds. *Left:* The learned step (ranging from 1 to 948) represents the step at which the implicit reward model distinguishes between preferred and rejected responses (see Eq. (2), threshold $\delta = 0.4$). X-axis: 40 unique combinations of model size (4 total) and training data subset (10 per model). Y-axis: 300 test examples, sorted by average learned step across 40 runs. Color gradients encodes difficulty. *Middle:* Two *Spearman's* rank correlation matrices. Lower triangle: correlations of learned step across runs; upper triangle: validation loss correlations. *Right:* Two *Jaccard* similarity matrices for difficult examples (top 50%) defined by learned step and validation loss across runs.

# 3. Preference Examples Vary in Difficulty

Examples are learned in a remarkably consistent order, revealing the inherent example difficulty. We then validate the validation loss as an effective measure of this difficulty for alignment tasks.

## 3.1. The Underlying Example Difficulty

While various metrics such as length (Spitkovsky et al., 2010; Tay et al., 2019; Nagatsuka et al., 2023) and perplexity (Wu et al., 2024) have been proposed to measure difficulty of text samples, their ability to reliably capture example difficulty remains controversial (Campos, 2021). We address this concern by demonstrating: (1) examples have distinct learned steps (see Eq.2), indicating different difficulty levels, and (2) these learned steps are consistent across runs with different training data and random seeds.

In Figure 2 (left), we visualize the learned steps of 300 test examples from *Ultrafeedback-binarized*[2],where darker colors indicate more training steps needed for model comprehension. Results from 10 runs show consistent learning order across different models (Jiang et al., 2023; AI@Meta, 2024; Team et al., 2024) varying in size (2B–9B), training stage, and data sampling. This consistency confirms that examples vary in difficulty, allowing us to discuss difficult examples without debating various definitions of difficulty.

## 3.2. Validation Loss as a Proxy for Learned Step

The robust learning order suggests the existence of difficult examples—some examples are consistently harder for LLMs to understand. However, identifying these examples

---

[2] https://huggingface.co/datasets/HuggingF aceH4/ultrafeedback_binarized

at scale is computationally expensive, as the computing of learned step requires evaluating the model after each gradient update. To address this, we adopt the validation loss from the curriculum learning literature (Wu et al., 2021; Rampp et al., 2024) (see Eq(3)). Specifically, we train six reference models using the DPO objective on the randomly sampled half training set and evaluate the validation loss for examples on the other half. We refer the difficult examples to examples with a large validation loss.

**Definition 3.1** (Difficult example). A preference example $(x, y_w, y_l)$ is considered a *difficult example* if its *validation loss* is no less than a specified value:

$$\text{VL}(x, y_w, y_l) \geq Q(\tau).$$

*Remark* 3.2. We introduce a flexible threshold $Q(\tau)$ which is the $\tau$-quantile of the validation loss. This accounts for the lack of a formal definition of sample difficulty (Zhu et al., 2024b) and the variation in loss distributions across models.

To assess whether the validation loss effectively approximates the learned step, we examine the correlation between difficulty rankings produced by these two measures. Using *Spearman's* rank correlation, we compared rankings across different runs and models. As shown in the middle panel of Figure 2, the validation loss exhibits patterns remarkably similar to the learned step. Furthermore, the high correlation coefficients between average learned step and average validation loss across the four models (0.9258, 0.9227, 0.9336, and 0.9283) validate the effectiveness of validation loss as a computationally efficient proxy for learned step. Additionally, the *Jaccard similarity* between difficult example sets (defined as top 50% by either metric) remains consistently high for each model (Figure 2, right), confirming that both measures identify similar sets of difficult examples.

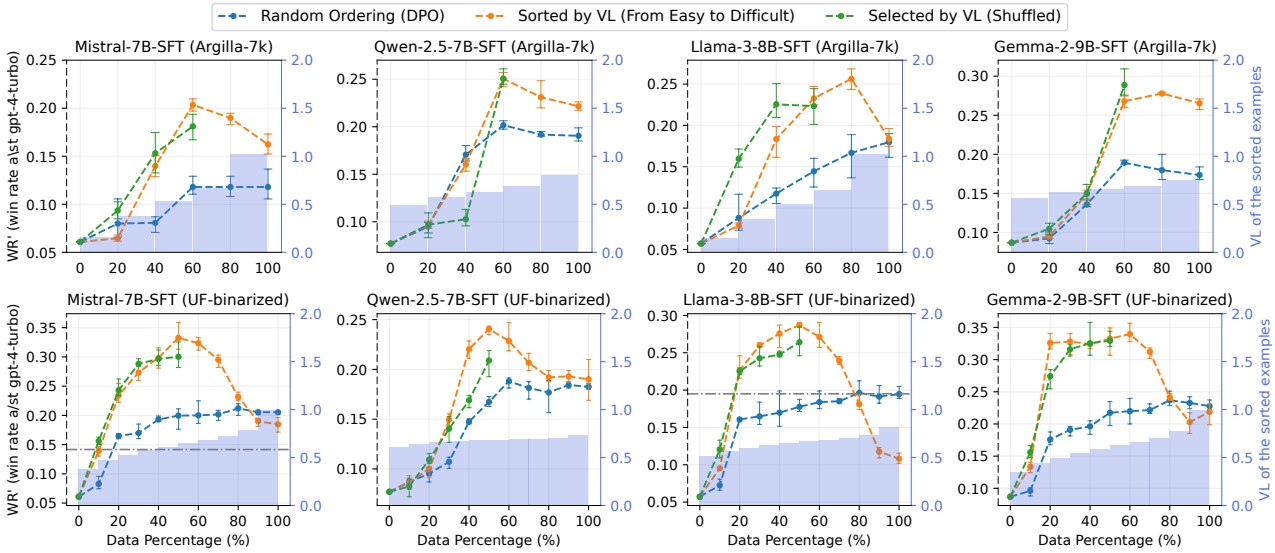

Figure 3: **Direct Preference Optimization (DPO) struggles with difficult examples, broadly and significantly.** We present the defined WR′ evolution for four models trained on the *argilla-mix-dpo-7k* and *ultrafeedback-binarized* datasets. The results are based on checkpoints from three 1-eopch runs with different seeds. *Random Ordering (DPO)*: Training data are presented in a randomized sequence. *Sorted by VL (From Easy to Difficult)*: Training examples are ranked by their *validation loss* (VL) and presented from easy to difficult, following a curriculum learning approach. *Selected by VL (Shuffled)*: The easiest 60% (for Argilla-7K) or 50% (for UF-binarized) of the data is selected based on VL, and examples are sampled in a random order for training. The VL measurements are displayed as bar plots. We include evaluation results (dashed lines) from the two corresponding DPO models released by Meng et al. (2024) for reference.

## 4. Difficult Examples Hinder Alignment

In this section, we first demonstrate that difficult examples significantly degrade alignment performance across various datasets and model scales. We then investigate the factors that contribute to their difficulty through a series of systematically designed empirical studies.

### 4.1. Investigation Setup

**Models.** We start the alignment from SFT models trained on the *UltraChat-200k* dataset: **Mistral-7B-SFT** (Jiang et al., 2023), **Qwen-2.5-7B-SFT** (Yang et al., 2024), **Llama3-8B-SFT** (AI@Meta, 2024), and **Gemma-2-9B-SFT** (Team et al., 2024). This setting better demonstrates the effects of different alignment procedures (Meng et al., 2024).

**Datasets.** We use *UltraFeedback-binarized*, a widely adopted alignment dataset (Tunstall et al., 2023; Meng et al., 2024; Zhou et al., 2024; Pattnaik et al., 2024), and *Argilla-dpo-mix-7k*[3], a small but high-quality dataset.

**Hyper-parameters.** Following prior work, we set $\beta = 0.01$ (Zhou et al., 2024). The learning rate is swept for DPO with random ordering and directly applied to DPO with other settings. We conduct the alignment with one epoch

[3]https://huggingface.co/datasets/argilla/dpo-mix-7k

following Meng et al. (2024).

**Evaluation.** We employ **WR′**, the win rate against *gpt-4-turbo* on 805 testing examples from *AlpacaEval 2* (Dubois et al., 2024) with *ArmoRM* (Wang et al., 2024c), a reward model with impressive performance on the Reward-Bench (Lambert et al., 2025), as the evaluator. This evaluation setup allows us to evaluate thousands of checkpoints.

### 4.2. Difficult Examples Hinder Preference Alignment

As shown in Figure 3, training on difficult examples leads to significant performance declines. We compare three example-ordering strategies: (1) random ordering (standard DPO), (2) easy-to-difficult sorting by validation loss, and (3) random ordering with only easy examples. Despite using the same training recipes, models consistently perform better when trained on easier examples across all four architectures and both datasets. Notably, the benefits are mainly unlocked by excluding difficult examples rather than the ordering itself, as shown by the similar performance of sorted and shuffled easy examples (Strategies 2 and 3).

The performance drop due to difficult examples is more pronounced in *Ultrafeedback-binarized*. This is aligned with the observation that *Ultrafeedback-binarized* contains mislabeled examples (Argilla, 2024; Bartolome et al., 2023) and *Argilla-dpo-mix-7k* is characterized by high-quality data.

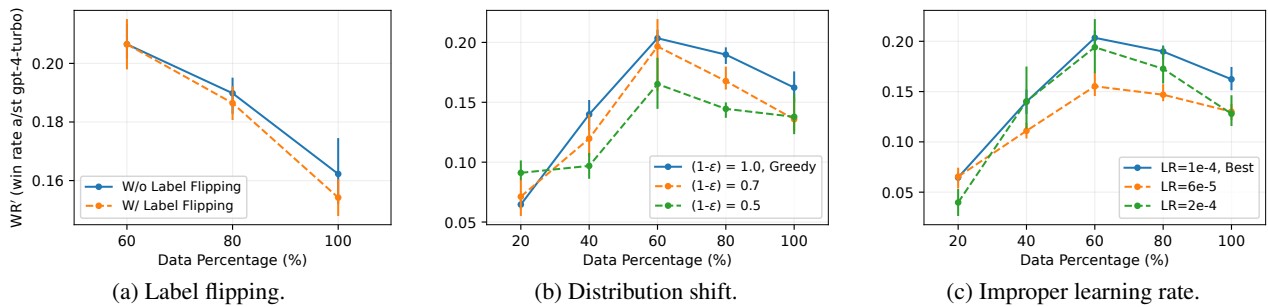

(a) Label flipping.    (b) Distribution shift.    (c) Improper learning rate.

Figure 4: **Difficulty examples are not necessarily data errors.** *(a)*: flipping the last 40% examples with higher *validation loss*. *(b)*: sorting the examples with the $\epsilon$-*greedy sorting* algorithm. In this case, each mini-batch data contains $(1-\epsilon)$ part of easy-to-difficult examples and $(\epsilon)$ part of randomly sampled examples. *(c)*: increasing and decreasing the learning rate. All experiments are conducted on the Mistral-7B-SFT model with *Argilla-dpo-mix-7k* dataset.

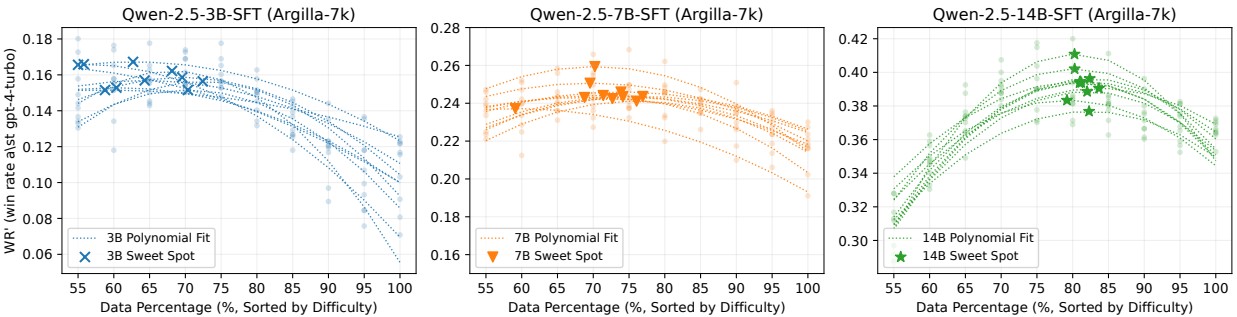

Figure 5: **Difficult examples benefit larger models with greater capacities.** Examples are sorted by their *validation loss*, ranging from easy to difficult. We fit the measured $\mathbf{WR'}$ (scatter points) using a second-degree polynomial (dashed line), identifying the peak of each parabola as the *sweet spot* (marker). Notably, larger models reach sweet spots at higher data percentages, indicating that model with greater capacity can manage more challenging examples. The results are from ten runs per model type, evaluated using *ArmoRM* (Wang et al., 2024c).

### 4.3. Difficult Examples Are Not Necessarily Data Errors

Before proposing our solution to filtering out difficult and harmful examples, we shed light on their traits to justify their removal here. For statistics and case study on difficult examples, please refer to Appendix E and F.

**Mislabeled data (Figure 4 (a)).** Prior work suggests that difficult examples might be mislabeled (Argilla, 2024; Bartolome et al., 2023). To test this hypothesis, we sort the examples by their validation loss and flip the labels of last 40% (the most difficult) examples. However, this modification does not alleviate the performance drop, suggesting that label noise is not the primary cause.

**Distribution shift (Figure 4 (b)).** Another possibility is that difficult examples represent a distinct distribution, causing catastrophic forgetting when models transition from easy to difficult examples. We test this using $\epsilon$-greedy sorting: each mini-batch contains $\epsilon$ portion of randomly sampled examples and $(1 - \epsilon)$ portion of examples sorted by validation loss. This ensures continuous exposure to both distributions, yet shows no improvement over the greedy sorting.

**Learning rate sensitivity (Figure 4 (c)).** We argue that the performance drop is not simply caused by the improper learning rate. We investigate this with a varying learning rate. However, adjusting the learning rate neither alleviates performance drops nor delays the decline, demonstrating that the issue is unrelated to improper optimization settings.

### 4.4. Difficult Example Exceeds Model's Capacity

We hypothesize that difficult examples bring about training tasks beyond the model's current capabilities, thus requiring larger models to properly understand the nuanced preference differences. To validate this hypothesis, we conduct experiments using Qwen-2.5 models (Yang et al., 2024) of three sizes: 3B, 7B, and 14B. The dataset is *Argilla-dpo-mix-7k*. Figure 5 shows a clear relationship between model size and manageable example difficulty: the optimal percentage of training data (the *sweet spot*) increases from 64% for the 3B model to 81% for the 14B model. This scaling pattern demonstrates that larger models can effectively learn from more difficult examples, confirming the direct relationship between model capacity and example difficulty threshold.



Figure 6: **The pipeline of *Selective DPO*.** It extends DPO (Rafailov et al., 2024) with a principled data selection process: selecting preference examples within the model's capacity. Specifically, Selective DPO comprises three steps: *(1)* Train a set of reference models using the DPO loss on different subsets of the training data. *(2)* Evaluate the reference models to compute the validation loss, which serves as a proxy for example difficulty. *(3)* Selectively align LLMs on examples with low validation loss from easy to difficult examples.

## 5. Selective DPO

Having verified the three key claims underpinning our data selection principle, we are now well-positioned to propose an instantiated algorithm, *Selective DPO*. It extends the standard DPO (Rafailov et al., 2024) by selectively training on examples within the model's capacity. The algorithm consists of three main steps, as illustrated in Figure 6:

- **Train reference models.** The training dataset is randomly split into two partitions. Using the standard DPO loss (Eq. 1), SFT models are trained separately on each partition, resulting in two reference models per split. This process is repeated three times, yielding six reference models. Unlike the reference SFT model used in the DPO objective to control KL divergence, these reference models are specifically employed for computing validation loss.

- **Rank examples by their validation loss.** The trained reference models evaluate held-out examples from their respective complementary partitions $(\mathcal{D}\backslash\hat{\mathcal{D}})$. Each example is assessed three times using different reference models, and the mean validation loss is computed to rank the examples in ascending order.

- **Align with the selected data.** The easiest examples, comprising the lowest $\tau$ percent of validation losses, are selected for alignment training. The alignment algorithm, such as DPO, is applied exclusively to these examples. To fully utilize the difficulty ranking, examples are processed sequentially from easy to difficult.

*Remark* 5.1 (Flexible hyper-parameter $\tau$). The optimal $\tau$, which determines the percentage of selected data, depends on the data difficulty distribution and the model's capacity. In practice, $\tau$ can be tuned using a third-party evaluator such as AlpacaEval 2 (Dubois et al., 2024). For the evaluation in the next section, we set $\tau = 50$ for the *UltraFeedback-binarized* dataset, based on insights from Figure 3. For clarity and reproducibility, pseudocode for Selective DPO is provided in Appendix A.

## 6. Experiments

We evaluate the proposed preference data selection principle by benchmarking the Selective DPO algorithm on formal benchmarks: *AlpacaEval 2* (Dubois et al., 2024), *Arena-Hard v0.1* (Li et al., 2024b), and *MT-Bench* (Zheng et al., 2023). We report scores following each benchmark's evaluation protocol.

### 6.1. Performance Comparison

**Baselines.** Data selection for alignment remains a relatively underexplored yet promising direction. To provide a comprehensive evaluation, we consider three categories of relevant baseline algorithms: (1) Data correction methods, including label flipping and label smoothing, aim to mitigate annotation errors; (2) DPO (Rafailov et al., 2024) and its variants such as IPO (Azar et al., 2024), KTO (Ethayarajh et al., 2024), ORPO (Hong et al., 2024), SimPO (Meng et al., 2024), and WPO (Zhou et al., 2024); (3) Potential solutions for preference data selection, such as CHES (Razin et al., 2025)—designed for refusal alignment on unsafe prompts—along with RM (filtering out samples with low reward margins (Gao et al., 2024)) and PPL (selecting SFT samples with moderate perplexity (Wu et al., 2024; Ji12 et al., 2024)). All baseline algorithms undergo hyperparameter tuning on the learning rate. Implementation details are provided in Appendix C.

**Results (Table 1 and Figure 7).** Table 1 compares results on the Mistral-7B (Jiang et al., 2023) and Llama-3-8B (AI@Meta, 2024) models. Label flipping yields only marginal gains, supporting our insight that difficult examples are not necessarily data errors. In contrast, Selective DPO, which carefully selects 50% of the training data, significantly outperforms all baselines across all three benchmarks, demonstrating the strength of our data selection principle for alignment tasks. Figure 7 extends the comparison to Gemma-2-9B (Team et al., 2024) and Qwen-2.5-7B (Yang et al., 2024), showing exceptional performance in win rate (WR) on AlpacaEval 2 and comparable performance on

Table 1: Benchmarking results from AlpacaEval 2 (Dubois et al., 2024), Arena-Hard (Li et al., 2024b), and MT-Bench (Zheng et al., 2023). In AlpacaEval 2, **WR** and **LC** indicate the win rate and length-controlled win rate against GPT-4-Turbo. We report the mean and standard variance across three runs. In Arena-Hard, **WR** represents the win rate against GPT-4-0314, with GPT-4-Turbo serving as the evaluator. MT-Bench scores the quality of generated responses on a scale from 1 to 10, using either GPT-4 or GPT-4-Turbo as the evaluator. All results are based on full parameter fine-tuning (FPFT), except for the row labeled with LoRA (Hu et al., 2022). We run this comparison on the *UltraFeedback-binarized* dataset.

| Method | Mistral-7B-SFT | | | | | Llama-3-8B-SFT | | | | |
| | AlpacaEval 2 | | Arena-Hard | MT-Bench | | AlpacaEval 2 | | Arena-Hard | MT-Bench | |
| | LC (%) | WR (%) | WR (%) | GPT-4 Turbo | GPT-4 | LC (%) | WR (%) | WR (%) | GPT-4 Turbo | GPT-4 |
|---|---|---|---|---|---|---|---|---|---|---|
| SFT | 8.4 | 6.2 | 1.3 | 4.8 | 6.3 | 6.2 | 4.6 | 3.3 | 5.2 | 6.6 |
| DPO (Rafailov et al., 2024) | 15.1 | 12.5 | 10.4 | 5.9 | 7.3 | 18.2 | 15.5 | 15.9 | 6.5 | 7.7 |
| + Label Flipping (Wang et al., 2024a) | 15.4 | 13.1 | 10.9 | - | 7.3 | 19.1 | 15.9 | 16.2 | - | 7.7 |
| + Label Smoothing (Mitchell, 2023) | 15.2 | 12.7 | 10.2 | - | 7.3 | 17.7 | 14.8 | 15.7 | - | 7.6 |
| RRHF (Yuan et al., 2023) | 11.6 | 10.2 | 5.8 | 5.4 | 6.7 | 12.1 | 10.1 | 6.3 | 5.8 | 7.0 |
| SLiC-HF (Zhao et al., 2023b) | 10.9 | 8.9 | 7.3 | 5.8 | **7.4** | 12.3 | 13.7 | 6.0 | 6.3 | 7.6 |
| IPO (Azar et al., 2024) | 11.8 | 9.4 | 7.5 | 5.5 | 7.2 | 14.4 | 14.2 | 17.8 | 6.5 | 7.4 |
| CPO (Xu et al., 2024) | 9.8 | 8.9 | 6.9 | 5.4 | 6.8 | 10.8 | 8.1 | 5.8 | 6.0 | 7.4 |
| KTO (Ethayarajh et al., 2024) | 13.1 | 9.1 | 5.6 | 5.4 | 7.0 | 14.2 | 12.4 | 12.5 | 6.3 | **7.8** |
| ORPO (Hong et al., 2024) | 14.7 | 12.2 | 7.0 | 5.8 | 7.3 | 12.2 | 10.6 | 10.8 | 6.1 | 7.6 |
| R-DPO (Park et al., 2024) | 17.4 | 12.8 | 8.0 | 5.9 | **7.4** | 17.6 | 14.4 | 17.2 | 6.6 | 7.5 |
| SimPO (Meng et al., 2024) | 21.5 | 20.8 | 16.6 | 6.0 | 7.3 | 22.0 | 20.3 | **23.4** | 6.6 | 7.7 |
| WPO (Zhou et al., 2024) | 24.4 | 23.7 | **16.7** | - | **7.4** | **23.1** | **22.2** | 23.1 | - | 7.7 |
| CHES(lowest 50%) (Razin et al., 2025) | $18.9_{0.74}$ | $16.6_{1.13}$ | - | - | - | $17.1_{0.69}$ | $15.9_{1.11}$ | - | - | - |
| RM(highest 50%) (Gao et al., 2024) | $16.2_{0.66}$ | $13.1_{1.21}$ | - | - | - | $19.7_{0.61}$ | $16.1_{1.24}$ | - | - | - |
| PPL(middle 50%) (Wu et al., 2024) | $17.3_{0.62}$ | $15.4_{1.10}$ | - | - | - | $15.3_{0.59}$ | $15.7_{1.10}$ | - | - | - |
| **Selective DPO** (Ours w/ LoRA) | $25.4_{0.80}$ | $27.4_{1.26}$ | 16.2 | - | 7.3 | $21.1_{0.73}$ | $18.3_{1.14}$ | 22.7 | - | **7.8** |
| **Selective DPO** (Ours) | $27.1_{0.63}$ | $28.9_{1.31}$ | **17.0** | - | 7.4 | $24.9_{0.77}$ | $25.3_{1.36}$ | 24.1 | - | 8.0 |

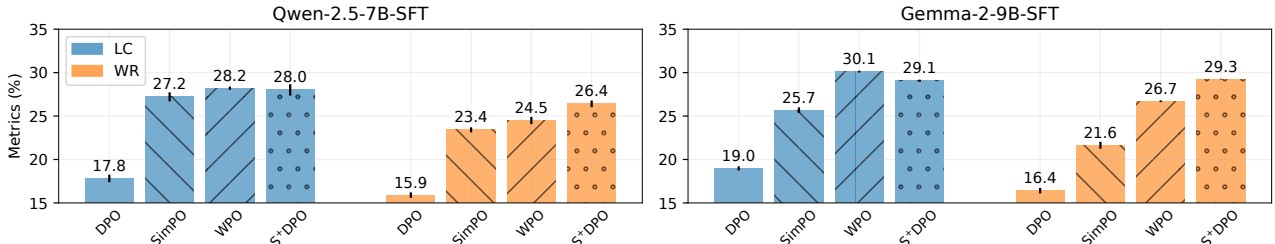

Figure 7: Comparison results against SimPO and WPO, with all methods tuned for their learning rates. Selective DPO (S$^+$DPO) demonstrates superior performance in win rate (WR) and comparable results in length-controlled win rate (LC).

length-controlled win rate (LC). The slightly lower performance on LC is consistent with results in Table 1, where Selective DPO demonstrates better performance under WR.

We emphasize that our goal is not to propose the best ever alignment algorithm, but to verify the proposed data selection principle for alignment: selecting examples that match the model's capacity. The length exploitation issue, while beyond the scope of this paper, could potentially be addressed using techniques from SimPO (Meng et al., 2024) or WPO (Zhou et al., 2024), which we leave as future work.

### 6.2. Hyper-Parameter Study

Selective DPO introduces two implicit hyperparameters. **Number of reference models (Figure 8 (a))**: Increasing

the number of reference models used to compute the validation loss improves performance on *AlpacaEval 2* (LC). However, considering computational costs, training six reference models strikes a balance between performance and efficiency. **Percentage of selected easy examples (Figure 8 (b))**: Increasing $\tau$ incorporates examples exceeding the model's capacity, leading to performance degradation, while excessively low values limit training to the simplest examples, also resulting in suboptimal performance.

### 6.3. In-Depth Analysis of DPO vs. Selective DPO

Selective DPO outperforms DPO in terms of likelihood distribution and reward margin distribution. As shown in Figure 8(c), Selective DPO achieves a distribution of negative log-likelihoods (NLLs) closer to zero on test prompts

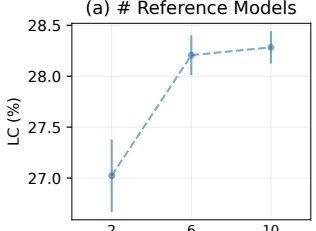 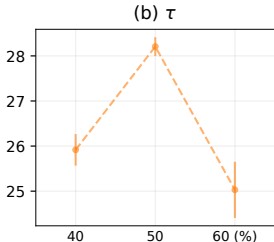 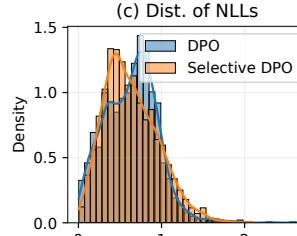 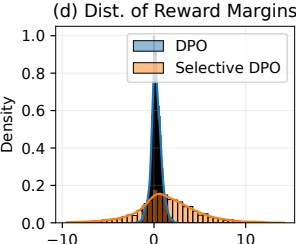

Figure 8: **Hyperparameter study and in-depth analysis of Selective DPO.** *(a)*: Relationship between the number of reference models and performance. *(b)*: Performance with different percentages of selected easy examples. *(c)*: Negative log-likelihoods distributions on the generated responses. *(d)*: Reward margin distributions of the implicit reward models.

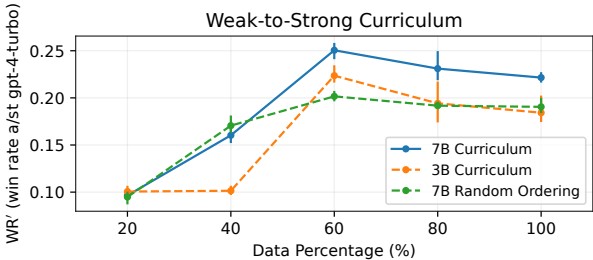

Figure 9: **Weak-to-strong curriculum under-performs.** Aligning a 7B model with examples ordered by 3B reference models yields compromised results.

from *AlpacaEval 2*, indicating higher confidence in generated responses. Additionally, the implicit reward model learned by Selective DPO exhibits better accuracy and larger reward margins on testing examples (Figure 8(d)).

### 6.4. Weak-to-Strong Curriculum

To investigate whether difficult examples can be identified using smaller reference models, we compare alignment experiments where a 7B SFT model is trained with its own curriculum versus a curriculum derived from a smaller 3B model. Results in Figure 9 show moderate benefits from the smaller model's curriculum, though slightly inferior to the model's own curriculum. This suggests that while smaller models can provide insights, data selection remains more effective when tailored to the target model's capacity.

### 7. Related Work

**Response selection.** The importance of selecting high-quality responses as preferred choices has been highlighted in several studies (Bai et al., 2022; Ethayarajh et al., 2022; Tunstall et al., 2023). These works focus on ensuring that preferred responses are aligned with human values. Our work builds upon these efforts in two key ways: (1) the datasets we consider already incorporate these response selection techniques, and (2) we prioritize whether preference examples fall within the capabilities of the target LLM, rather than solely emphasizing their alignment with human values. **Data correction.** Efforts to address noisy labels in-

clude techniques such as label flipping (Wang et al., 2024a) and confidence-based data filtering (Gao et al., 2024). Approaches like cDPO (Mitchell, 2023) and rDPO (Chowdhury et al., 2024) aim to mitigate the impact of mislabeling without explicitly removing mislabeled examples. In our study, we incorporate label flipping and label smoothing experiments to support our claim that difficult examples are not necessarily mislabeled examples, but rather those exceeding the model's capacity. **Seemingly relevant work.** Our study differs from general data selection research, such as Liu et al. (2024); Xia et al. (2024), which uses the term alignment but actually focuses on the SFT stage. For a comprehensive review of data selection for LLMs and curriculum learning, we refer readers to Appendix B.

### 8. Conclusion and Future Work

In this work, we reveal and address a critical gap in LLM alignment: the mismatch between data difficulty and model capacity. Challenging the assumption that more clean data uniformly improves alignment, we propose a novel principle for alignment tasks:

*Preference data vary in difficulty, and overly difficult examples hinder alignment, by exceeding the model's capacity.*

Comprehensive experiments validate the three key claims underlying this principle. Building on this data selection principle, we introduce Selective DPO, an alignment algorithm that selectively trains on examples within the model's capacity. Selective DPO achieves state-of-the-art results on benchmarks including AlpacaEval 2, Arena-Hard, and MT-Bench, with up to 16% gains in win rates over DPO. Our work advocates a paradigm shift in alignment: alignment should prioritize data difficulty relative to model capacity rather than treating all preference data equally.

However, limitations remain: (1) Selective DPO tends to favor longer responses due to potential data bias; and (2) the proposed principle is designed and validated specifically for the DPO setting, limiting its direct applicability to RLHF. These gaps highlight opportunities for future work.

## Acknowledgement

The authors thank Qichao Wang, Qingyang Zhang, and Ziqiao Meng for their valuable feedback during the initial phase of this work. The authors also thank Guangyi Chen, Yongqiang Chen, Guozheng Ma, Cong Zeng, Gongxu Luo, and Loka Li for their helpful discussions and suggestions during the writing phase. The authors are grateful to the open-source communities, especially the developers and researchers of alignment-handbook, SimPO and WPO, for generously sharing their data, results, and code.

## Impact Statement

This paper presents work whose goal is to advance the alignment between large language model behaviors and human values. There are many potential societal consequences of our work, none of which we feel must be specifically highlighted here.

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

# A. Pseudocode for the Instantiated Algorithm: Selective DPO

---

**Algorithm 1** Selective DPO
---

**Input:**

$\pi_{\text{SFT}}$: An SFT model that serves as the starting point for preference alignment.

$D$: A dataset consisting of preference examples.

*RandomSampler*: A utility for sampling elements randomly without replacement.

*SequentialSampler*: A utility for sampling elements sequentially.

$\mathcal{L}_{\text{DPO}}$: DPO loss function with the form: $\mathcal{L}_{\text{DPO}}(x, y_w, y_l) = -\log \sigma\left(\beta \log \frac{\pi_{\theta}(y_w|x)}{\pi_{\text{ref}}(y_w|x)} - \beta \log \frac{\pi_{\theta}(y_l|x)}{\pi_{\text{ref}}(y_l|x)}\right)$

\# Step 1: Train six reference alignment models: $\pi_{\theta_{01}}, \pi_{\theta_{02}}, \pi_{\theta_{11}}, \pi_{\theta_{12}}, \pi_{\theta_{21}}, \pi_{\theta_{22}}$.

**for** $t = 0, 1, 2$ **do**

    Randomly split the dataset $D$ into two subsets, $D_1$ and $D_2$.

    Initialize $\pi_{\text{ref}} \leftarrow \pi_{\text{SFT}}$ and $\pi_{\theta} \leftarrow \pi_{\text{SFT}}$.

    **while** *RandomSampler* has not finished **do**

        Sample a mini-batch of examples from $D_1$ using *RandomSampler*.

        Update $\pi_{\theta}$ by minimizing the DPO loss function: $\pi_{\theta} \leftarrow \arg\min_{\pi_{\theta}} \mathbb{E}_{(x, y_w, y_l) \sim D_1}\left[\mathcal{L}_{\text{DPO}}(x, y_w, y_l)\right]$

    **end while**

    Save the model: $\pi_{\theta_{t1}} \leftarrow \pi_{\theta}$.

    Reinitialize: $\pi_{\theta} \leftarrow \pi_{\text{SFT}}$.

    **while** *RandomSampler* has not finished **do**

        Sample a mini-batch of examples from $D_2$ using *RandomSampler*.

        Update $\pi_{\theta}$ by minimizing the DPO loss function: $\pi_{\theta} \leftarrow \arg\min_{\pi_{\theta}} \mathbb{E}_{(x, y_w, y_l) \sim D_2}\left[\mathcal{L}_{\text{DPO}}(x, y_w, y_l)\right]$

    **end while**

    Save the model: $\pi_{\theta_{t2}} \leftarrow \pi_{\theta}$.

    Reinitialize: $\pi_{\theta} \leftarrow \pi_{\text{SFT}}$.

**end for**

\# Step 2: Rank examples by their validation loss.

**for** each example $(x, y_w, y_l)$ in dataset $D$ **do**

    Compute the validation loss using the three held-out reference alignment models:

$$\text{VL}(x, y_w, y_l) = \mathbb{E}_{\pi_{\theta} \sim (\pi_{\theta_{01}} \text{ or } \pi_{\theta_{02}}, \pi_{\theta_{11}} \text{ or } \pi_{\theta_{12}}, \pi_{\theta_{21}} \text{ or } \pi_{\theta_{22}})}\left[\mathcal{L}_{\text{DPO}}(x, y_w, y_l)\right].$$

**end for**

Select the 50% examples with the lowest validation losses to form $D_{\text{selected}}$.

\# Step 3: Conduct alignment on the selected data $D_{\text{selected}}$.

**while** *SequentialSampler* has not finished **do**

    Sample a mini-batch of examples from $D_{\text{selected}}$ using *SequentialSampler*.

    Update $\pi_{\theta}$ by minimizing the DPO loss function: $\pi_{\theta} \leftarrow \arg\min_{\pi_{\theta}} \mathbb{E}_{(x, y_w, y_l) \sim D_{\text{selected}}}\left[\mathcal{L}_{\text{DPO}}(x, y_w, y_l)\right]$

**end while**

**Output:**

$\pi_{\theta}$: The aligned model obtained by Selective DPO.

---

# B. Related Work

## B.1. Data Selection for Pre-Training

Selecting training corpus brings significant performance gains in the pre-training stage (Wenzek et al., 2019; Brown et al., 2020; Zhao et al., 2023a; Penedo et al., 2023; Tang et al., 2024). Existing approaches can be broadly categorized into two categories: **Sample-level selection** focuses on filtering out undesired content such as non-target languages, duplicated data, toxic materials, and low-quality information (Albalak et al., 2024). This is often achieved through model-based filters (Joulin, 2016; Engstrom et al., 2024; Wettig et al., 2024) or heuristic filters (Wenzek et al., 2019; Lee et al., 2021; Laurençon et al., 2022), each applying specialized filters for specific objectives. **Token-level selection**, an emerging strategy, down-weights low-quality tokens to enhance data quality (Lin et al., 2024), complementing sample-level filtering.

## B.2. Data Selection for Supervised Fine-Tuning

Recent study suggests that SFT changes only the format of generation (Zhou et al., 2023). In light of this, various methods are proposed for finding the most informative subset for SFT, mainly following three principles: data quality, diversity, and importance (Qin et al., 2025). The measurement of **data quality** can be manual indicators such as the linguistic DQI (Mishra & Sachdeva, 2020), human scores (Zhou et al., 2023). Model-based quality measurement includes predictions from ChatGPT (Chen et al., 2024a), reward models (Cao et al., 2023), small reference models (Ankner et al., 2024) and the LLM itself (Li et al., 2024a). Measurements of **data diversity** are mainly manually defined, such as the source diversity (Mukherjee et al., 2023; Wang et al., 2023) and distance in the embedding space (Wu et al., 2023; Xu et al., 2023; Du et al., 2023; Chen et al., 2024c; Liu et al., 2024). **Data importance**, which evaluates an example's contribution to a specific task, measured using performance scores (Engstrom et al., 2024), data influence models (Yu et al., 2024), or relevance to desired skills (Chen et al., 2024b).

## B.3. Scoring the Example Difficulty

Scoring data difficulty is central to curriculum learning, which prioritizes training on simpler examples before progressing to more complex ones (Bengio et al., 2009). **Heuristic scoring functions** mirror human priors of difficulty understanding, such as sentence length (Spitkovsky et al., 2010; Tay et al., 2019; Nagatsuka et al., 2023), word rarity (Chang et al., 2021), and linguistic perplexity (Campos, 2021). In contrast, **principled scoring functions** leverage model behavior to indicate example difficulty, including reward margins from third-party reward models (Croitoru et al., 2024), model perplexity on responses (Wu et al., 2024), attention patterns (Ghosal et al., 2024) or attention scores from transformer models (Kim & Lee, 2024). In addition, we refer readers interested in training dynamics—the motivation behind our work—to Shen & Sanghavi (2019), Liu (2021), and Swayamdipta et al. (2020) for further insights into learning dynamics and sample difficulty. In this work, we employ two principled scoring measures, demonstrating their robustness and consistency in ranking examples. This allows us to analyze difficult examples objectively, avoiding ambiguities inherent in heuristic definitions.

## B.4. Curriculum Learning for Alignment

Curriculum learning (CL) mimics human cognition by structuring learning from simpler to more complex concepts (Avrahami et al., 1997; Bengio et al., 2009). However, CL remains a highly debated technique. While some studies show that it accelerates convergence, enhances generalization, and/or improves robustness in models like convolutional neural networks (Jiang et al., 2014; Tudor Ionescu et al., 2016), recurrent neural networks (Zaremba & Sutskever, 2014; Sachan & Xing, 2016), transformers (Platanios et al., 2019), and diffusion models (Croitoru et al., 2023), other research finds little or no benefit (Platanios et al., 2019; Campos, 2021; Wu et al., 2021). In preference alignment for LLMs (Rafailov et al., 2024; Wang et al., 2024b), the results are similarly mixed. Kim & Lee (2024) explored CL for preference alignment and concluded that sorting examples according to *prompt length* and *attention score* offered no clear benefits. On the other hand, Pattnaik et al. (2024) reported positive results, albeit with other tricks such as multiple candidate pairs data and iterative reference model. Our study suggests that CL, when paired with robust difficulty scoring, can positively impact LLM alignment by aligning data difficulty with model capacity.

# C. Experiment Details

## C.1. Computational Environment

All training experiments in this paper were conducted on compute nodes equipped with $8 \times$ H100 GPUs. To facilitate reproduction with limited computational resources, we also provide key benchmarking results for selected models trained using $4 \times$ A100 40G GPUs with LoRA. Reproducing our SelectiveDPO on 7B models takes about 8 GPU hours (H100).

## C.2. SFT Hyper-Parameters

In this work, we limited our alignment experiments to SFT models, which is expected to better demonstrate the effects of different preference alignment procedures. We prepared these SFT models using the the UltraChat-200k dataset. We try our best to use the SFT models from community to facilitate the reproduction. However, there were no available SFT checkpoints for some pre-trained models (*e.g.,* Qwen-2.5 models). We in this part list the hyper-parameters for training these community-released SFT models as well as the SFT models trained by ourselves in Table 2.

Table 2: Training recipes for SFT models used in our experiments.

| SFT Model Name | Base Model Name | Batch Size | Learning Rate | Epoch | Optimizer | LoRA? |
|---|---|---|---|---|---|---|
| Qwen-2.5-3B-SFT | Qwen/Qwen2.5-3B | 128 | 2e-5 | 1 | Adam | No |
| Qwen-2.5-7B-SFT | Qwen/Qwen2.5-7B | 128 | 1e-5 | 1 | Adam | No |
| Qwen-2.5-14B-SFT | Qwen/Qwen2.5-14B | 128 | 5e-6 | 1 | Adam | No |
| Mistral-7B-SFT (HuggingFaceH4/mistral-7b-sft-beta) | mistralai/Mistral-7B-v0.1 | 128 | 2e-5 | 1 | Adam | No |
| Llama-3-8B-SFT (princeton-nlp/Llama-3-Base-8B-SFT) | meta-llama/Meta-Llama-3-8B | 128 | 2e-5 | 1 | Adam | No |
| Gemma-2-9B-SFT (tanliboy/zephyr-gemma-2-9b-sft) | google/gemma-2-9b | 128 | 3e-6 | 1 | Adam | No |

## C.3. Key Hyper-Parameters for Alignment

**Figure 3** We conducted a series of alignment experiments with LoRA on two datasets for generating Figure 3. Key hyperparameters used in the *Argilla-dpo-mix-7k* experiments are listed in Table 3 where we report the sweep range and the selected best learning rate for DPO in bold font. These parameters are then directly applied to other two settings (sorted and selected by VL) for generating Figure 3. The key parameters used for the UltraFeedback-binarized dataset are list in Table 4.

Table 3: Key hyperparameters used for aligning models on the **argilla-7k** dataset: Figure 3, top.

| Model for Alignment | Learning Rate | Batch Size | $\beta$ | Epoch | Optimizer | LoRA? |
|---|---|---|---|---|---|---|
| Mistral-7B-SFT | 2e-5, 3e-5, 5e-5, **1e-4**, 2e-4 | 64 | 0.01 | 1 | paged_adamw_32bit | Yes |
| Qwen-2.5-7B-SFT | 2e-5, 3e-5, **5e-5**, 1e-4, 2e-4 | 64 | 0.01 | 1 | paged_adamw_32bit | Yes |
| Llama-3-8B-SFT | 5e-5, 1e-4, 2e-4, **3e-4**, 5e-4 | 64 | 0.01 | 1 | paged_adamw_32bit | Yes |
| Gemma-2-9B-SFT | 1e-5, **2e-5**, 3e-5, 5e-5, 1e-4 | 64 | 0.01 | 1 | paged_adamw_32bit | Yes |

Table 4: Key hyperparameters used for aligning models on the **ultrafeedback-bianrized** dataset: Figure 3, bottom.

| Model for Alignment | Learning Rate | Batch Size | $\beta$ | Epoch | Optimizer | LoRA? |
|---|---|---|---|---|---|---|
| Mistral-7B-SFT | 1e-6, 3e-6, 5e-6, **8e-6**, 10e-6 | 64 | 0.01 | 1 | paged_adamw_32bit | Yes |
| Qwen-2.5-7B-SFT | 1e-6, **3e-6**, 5e-6, 8e-6, 10e-6 | 64 | 0.01 | 1 | paged_adamw_32bit | Yes |
| Llama-3-8B-SFT | 1e-6, 3e-6, 5e-6, 8e-6, **10e-6** | 64 | 0.01 | 1 | paged_adamw_32bit | Yes |
| Gemma-2-9B-SFT | 1e-6, 3e-6, **5e-6**, 8e-6, 10e-6 | 64 | 0.01 | 1 | paged_adamw_32bit | Yes |

**Table 1** Comparison results of this table are mainly borrowed from the SimPO paper (Meng et al., 2024). All results are obtained by full parameter fine-tuning (FPFT) except for the row labeled with LoRA. We added the results of our *Selective DPO* pipeline using the configurations detailed in the following table. The inclusion of LoRA results is to facilitate the reproduction for practices with limited resources.

Table 5: Key hyperparameters used for aligning models on the **ultrafeedback-bianrized** dataset: Figure 3, bottom.

| Experiment Name | Learning Rate | Batch Size | $\beta$ | Epoch | Optimizer | LoRA? |
|---|---|---|---|---|---|---|
| SelectiveDPO & Mistral-7B-SFT & LoRA | 8e-6 | 64 | 0.01 | 1 | paged_adamw_32bit | Yes |
| SelectiveDPO & Mistral-7B-SFT | 2e-7, 5e-7, **1e-6**, 2e-6, 3e-6 | 128 | 0.01 | 1 | paged_adamw_32bit | No |
| SelectiveDPO & Llama-3-8B-SFT & LoRA | 10e-6 | 64 | 0.01 | 1 | paged_adamw_32bit | Yes |
| SelectiveDPO & Llama-3-8B-SFT | 2e-7, 5e-7, **1e-6**, 2e-6, 3e-6 | 128 | 0.01 | 1 | paged_adamw_32bit | No |
| WPO & Llama-3-8B-SFT | 5e-7, **1e-6**, 2e-6 | 128 | 0.01 | 1 | paged_adamw_32bit | No |

**Figure 7** Comparison results of this figure are from runs with full parameter fine-tuning. We rerun two state-of-the-art alignment algorithms, SimPO (Meng et al., 2024) and WPO (Zhou et al., 2024) with hyperparameter sweeping on the learning rate. Other hyperparameter configurations follow the suggestion from their papers. Specifically:

Table 6: Key hyperparameters used for generating comparison in Figure 7.

| Experiment Name | Learning Rate | Batch Size | Epoch | Optimizer | Other Hyper-Parameters | LoRA? |
|---|---|---|---|---|---|---|
| WPO & Qwen-7B-SFT | 5e-7, **1e-6**, 2e-6, | 128 | 1 | paged_adamw_32bit | $\beta = 0.01$ | No |
| WPO & Gemma-9B-SFT | 2e-7, **5e-7**, 1e-6 | 128 | 1 | paged_adamw_32bit | $\beta = 0.01$ | No |
| SimPO & Qwen-7B-SFT | 6e-5, **8e-6**, 1e-5 | 128 | 1 | paged_adamw_32bit | $\beta = 2, \frac{\gamma}{\beta} = 0.8$ | No |
| SimPO & Gemma-9B-SFT | 5e-7, **1e-6**, 2e-6 | 128 | 1 | paged_adamw_32bit | $\beta = 2, \frac{\gamma}{\beta} = 0.8$ | No |
| SelectiveDPO & Qwen-7B-SFT | 5e-7, **8e-7**, 1e-6 | 128 | 1 | paged_adamw_32bit | $\beta = 0.01$ | No |
| SelectiveDPO & Gemma-9B-SFT | 2e-7, **3e-7**, 5e-7 | 128 | 1 | paged_adamw_32bit | $\beta = 0.01$ | No |

## C.4. LoRA Configuration for Alignment

We conduct all our analytics experiments using LoRA. Its detailed configurations are described in Table 7.

Table 7: LoRA configuration for all analytics experiments.

| Parameter | Value |
|---|---|
| load_in_4bit | false |
| lora_r | 16 |
| lora_alpha | 16 |
| lora_dropout | 0.05 |
| lora_target_modules | q_proj,k_proj,v_proj,o_proj,gate_proj,up_proj,down_proj |

## C.5. Decoding Configuration

**AlpacaEval 2.** For this benchmark, we employ sampling-based decoding strategies, configuring the temperature as follows: 0.7 for Mistral models, 0.9 for Llama-3 models, 0.5 for Gemma-2 models, and 0.7 for Qwen-2.5 models. These configurations are aligned with standard practices in the community.

**Arena-Hard.** For this benchmark, we utilize default greedy decoding across all settings, as outlined in Meng et al. (2024).

**MT-Bench.** We adapt the official decoding configuration, which varies in sampling temperatures for different models.

# D. Downstream Task Evaluation

To examine how the proposed selective preference optimization pipeline affects downstream task performance, we evaluate the instantiated algorithm, *Selective DPO*, alongside other baseline algorithms on various tasks listed in the HuggingFace Open Leaderboard (Beeching et al., 2023). Results, following established evaluation protocols, are presented in Table 8.

Table 8: Downstream task evaluation results. The dataset is *UltraFeedback-binarized*.

| | MMLU(5) | Winograd(5) | GSM8K(5) | HellaSwag(10) | ARC(25) | TruthfulQA(0) | Average |
|---|---|---|---|---|---|---|---|
| **Mistral-7B-Base** | | | | | | | |
| Base | 62.46 | 78.93 | 38.29 | 83.38 | 61.6 | 42.64 | 61.22 |
| SFT | 59.77 | 77.58 | 40.71 | 82.28 | 58.19 | 43.05 | 60.26 |
| DPO | 57.38 | 77.35 | 30.4 | 83.58 | 61.18 | 53.11 | 60.50 |
| SimPO | 58.43 | 77.35 | 32.3 | 83.54 | 61.95 | 50.82 | 60.73 |
| WPO | 59.54 | 78.69 | 32.07 | 85.23 | 64.08 | 51.04 | 61.78 |
| SelectiveDPO | 59.34 | 76.16 | 14.48 | 83.25 | 65.27 | 51.95 | 58.41 |
| SelectiveDPO(60%) | 59.54 | 76.87 | 28.58 | 84.25 | 65.96 | 57.21 | 62.07 |
| **Qwen-2.5-7B-Base** | | | | | | | |
| Base | 74.16 | 76.72 | 82.18 | 80.03 | 63.23 | 56.38 | 72.12 |
| SFT | 73.86 | 75.77 | 81.43 | 80.71 | 62.71 | 55.67 | 71.69 |
| DPO | 74.06 | 75.61 | 82.79 | 81.73 | 65.70 | 60.92 | 73.47 |
| SimPO | 74.33 | 77.11 | 85.22 | 82.48 | 68.09 | 65.51 | 75.45 |
| WPO | 74.29 | 75.85 | 83.55 | 83.2 | 68.52 | 65.09 | 75.08 |
| SelectiveDPO | 74.05 | 75.85 | 80.44 | 82.82 | 67.32 | 63.80 | 74.04 |
| **Llama-3-8B-Base** | | | | | | | |
| Base | 65.14 | 76.64 | 48.45 | 81.88 | 58.87 | 43.93 | 62.49 |
| SFT | 63.79 | 76.64 | 50.57 | 81.40 | 60.84 | 45.33 | 63.10 |
| DPO | 63.47 | 76.95 | 54.81 | 83.71 | 64.51 | 53.45 | 66.15 |
| SimPO | 63.18 | 77.58 | 47.76 | 82.93 | 65.44 | 59.44 | 66.06 |
| WPO | 63.46 | 76.72 | 44.58 | 84.14 | 65.27 | 53.84 | 64.67 |
| SelectiveDPO | 63.99 | 76.48 | 48.75 | 83.51 | 64.93 | 51.34 | 64.83 |
| **Gemma-2-9B-Base** | | | | | | | |
| Base | 70.29 | 80.03 | 40.41 | 82.66 | 67.83 | 45.56 | 64.46 |
| SFT | 70.82 | 78.77 | 41.93 | 83.53 | 68.77 | 48.04 | 65.31 |
| DPO | 71.17 | 80.11 | 44.43 | 85.42 | 71.33 | 56.96 | 68.24 |
| SimPO | 72.16 | 80.43 | 42.53 | 86.06 | 73.12 | 65.34 | 69.94 |
| WPO | 70.88 | 79.40 | 43.14 | 85.64 | 70.99 | 53.44 | 67.25 |
| SelectiveDPO | 70.88 | 79.56 | 43.67 | 85.30 | 70.82 | 54.67 | 67.48 |

Overall, *Selective DPO* performs comparably to other alignment algorithms, such as DPO and SimPO. However, we observe a notable performance drop in the Mistral-7B model when evaluated using the GSM8K (Cobbe et al., 2021) protocol. Two primary factors contribute to this performance decrease: **Exclusion of difficult examples.** GSM8K predominantly evaluates mathematical skills, which often correspond to difficult examples (as detailed in Appendix F). Since *Selective DPO* excludes such difficult examples to get better aligned with human preferences, the model's mathematical performance diminishes. **Formatting requirements**. GSM8K requires numerical answers in a specific format: ### <THE ANSWER>. We find that the Mistral-7B-Selective DPO model often generates correct answers but presents them in a human dialogue style, breaking the required format and reducing evaluation scores.

We propose three potential solutions. First, as suggested by SimPO (Meng et al., 2024), incorporating an auxiliary SFT loss to regularize model behavior could help regularize the model's behavior, ensuring compatibility with downstream tasks. Second, using larger models with greater capacity mitigates this issue. For instance, Gemma-2-9B-SelectiveDPO demonstrates better performance and is unaffected by this issue. Finally, including more examples that cover mathematical questions could prevent the model from forgetting its mathematical capabilities while aligned with human preferences and dialogue styles. For example Selective DPO(60%), which incorporates 10% more difficult data, alleviates this issue.

# E. What Defines Difficult Examples: Insights from Feature Analysis

## E.1. Can Length and Reward Margin Predict Example Difficulty?

We include alternative measures that could potentially indicate example difficulty and evaluate their behavior across varying levels of difficulty.

**Response length.** Response length may implicitly signal the complexity of generated answers, as longer responses often carry more information, potentially making them more challenging for the model. Two measures are defined: (1) *Chosen Length*: The length of the chosen answer, $\text{len}(y_w)$; (2) *Chosen Length − Rejected Length*: The difference in lengths between the chosen and rejected answers: $\text{len}(y_w) - \text{len}(y_l)$.

**Reward margin by reward models.** Reward models, such as *ArmoRM* (Wang et al., 2024c), provide score margins that can indicate response difficulty. A large positive margin suggests an easy example, while a large negative margin may signal noisy or mislabeled data. Two measures are defined: (1) *Chosen Score*: The reward score assigned to the chosen answer, $\text{rm}(x, y_w)$, and (2) *Chosen Score - Rejected Score*: The difference in scores between the chosen and rejected answers: $\text{rm}(x, y_w) - \text{rm}(x, y_l)$.

**Reward margin by GPT-4.** GPT-4 can also act as an evaluator, assigning scores to responses. Similar measures are defined: (1) *Chosen Rating*: The rating assigned to the chosen answer, $\text{GPT-4}(x, y_w)$, and (2) *Chosen Rating - Rejected Rating*: The difference in ratings between the chosen and rejected answers: $\text{GPT-4}(x, y_w) - \text{GPT-4}(x, y_l)$.

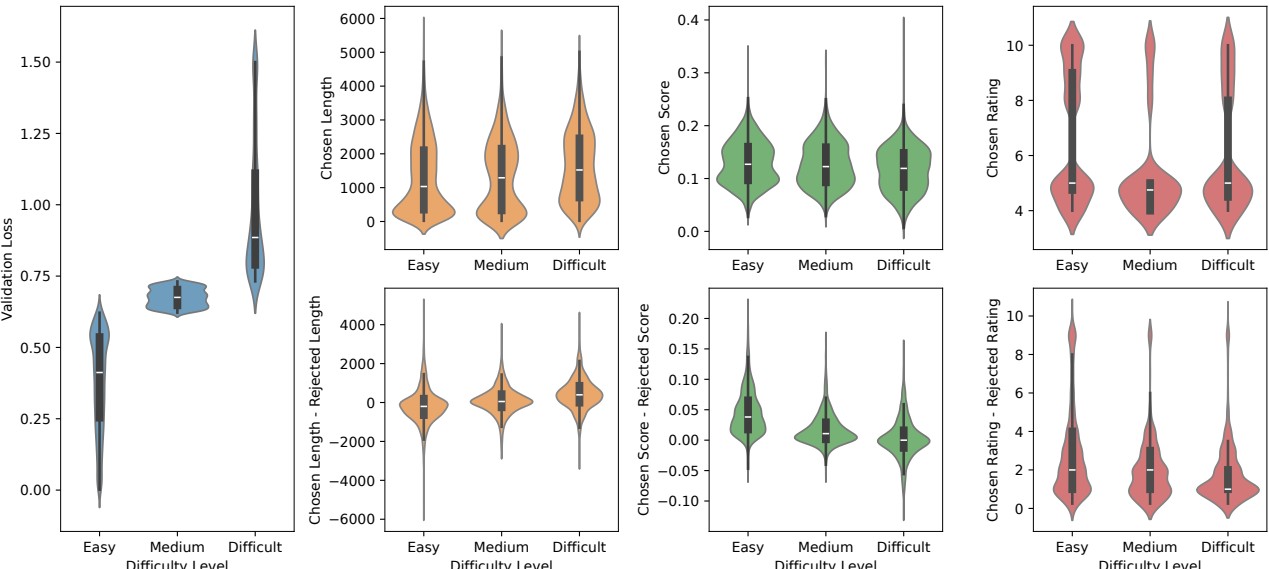

Figure 10: Comparison of *response length* and *reward margin* measures with *validation loss* across three difficulty levels: Easy, Medium, and Difficult. The dataset examples are partitioned into these levels based on increasing *validation loss*. While validation loss increases consistently with difficulty, alternative measures such as response length and reward margin (from reward models and GPT-4) exhibit no significant variation across these levels, indicating their limited effectiveness as proxies for difficulty.

**Comparison.** The distributions of these measures are shown in Figure 10. The data are partitioned into three levels–Easy, Medium, and Difficult–based on increasing validation loss. Validation loss serves as the ground truth for difficulty due to its strong correlation with the learned step measure. Difficult examples tend to have longer responses and larger negative reward margins. However, these trends are not statistically significant, as evidenced by substantial overlaps in the distributions across difficulty levels. This suggests that while these measures provide some insight, they may not serve as robust standalone indicators of example difficulty.

### E.2. Are Easy Examples for Small Models Still Easy for Larger Models?

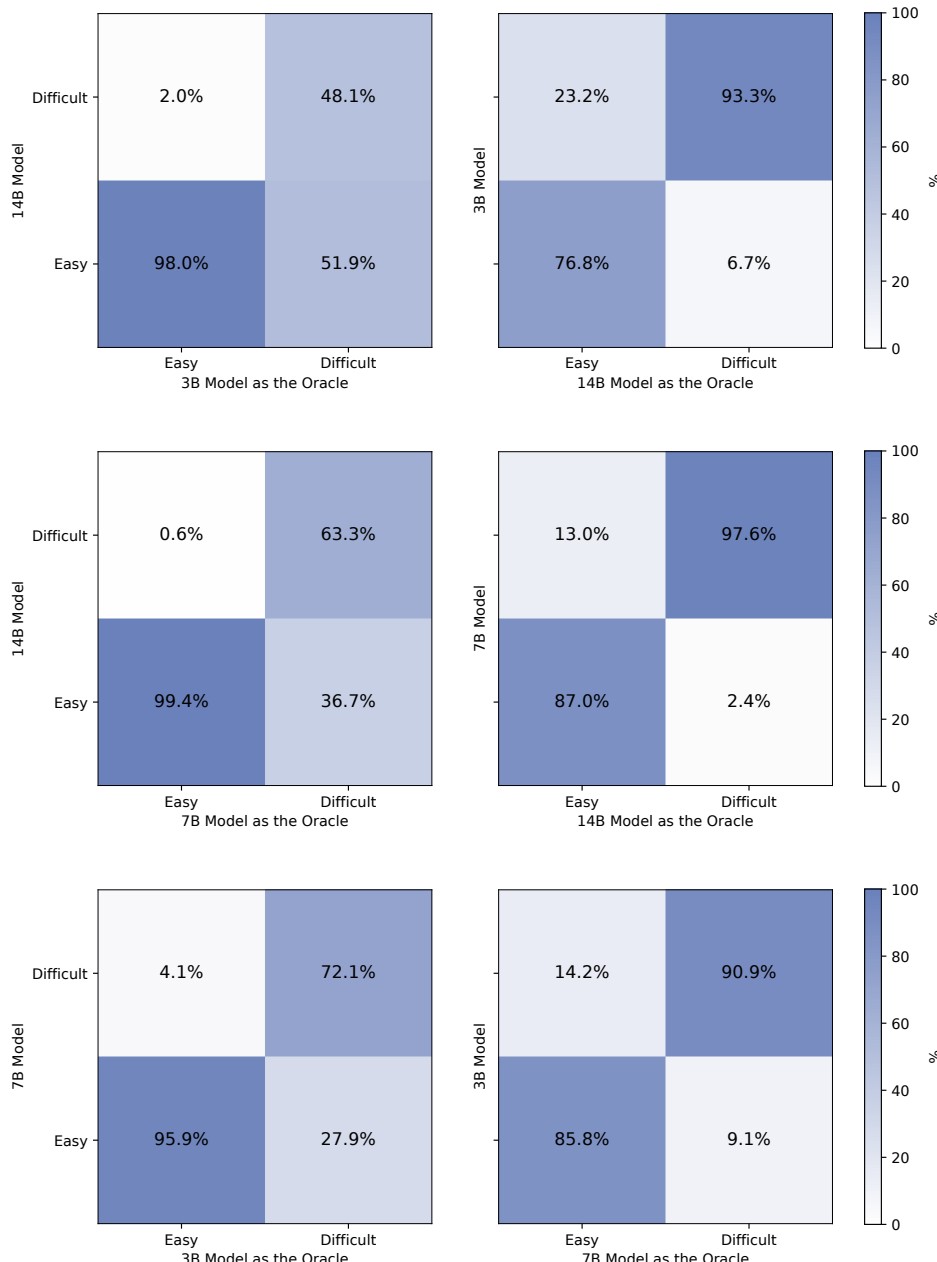

Figure 11: Easy examples identified by smaller models are likely also recognized as easy by larger models, and difficult examples identified by larger models are likely challenging for smaller models as well. Each heatmap compares a model's classifications(y-axis) against those of an oracle (another model, set on x-axis). Easy examples are defined as the first 63.7%, 71.2%, and 81.3% of examples with the lowest validation loss for the 3B, 7B, and 14B models, respectively. Difficult examples comprise the remaining examples, with these thresholds drawn from the mean sweet spot in Figure 5. The dataset is *Argilla-dpo-mix-7k*.

# F. What Defines Difficult Examples: Insights from Case Studies

In this section, we examine examples with varying levels of difficulty, characterized by their validation losses. We select three representative examples around validation loss quantile: 0.0 (easiest), 0.6 (medium), and 1.0 (most difficult) from the *Argilla-dpo-mix-7k* dataset. We highlight examples near the 0.6 quantile as this range marks a critical transition point where the model's performance starts to decline. Our key observations are as follows:

- **Easy examples are well-labeled and involve straightforward concepts (Table 9).** Easy examples show that preferred responses are concise, clear, and of significantly higher quality than their rejected counterparts, which are often overly verbose and filled with excessive emojis. Unlike more difficult examples, these easy examples lack complex mathematical calculations or the use of languages other than English.

Table 9: Three examples with the lowest validation loss (Model: Mistral-7B-SFT & Dataset: Argilla-dpo-7k).

| Prompt | Chosen | Rejected |
|---|---|---|
| How can you determine the pH of rainwater? How can you determine the pH level of rainwater? Choose your answer. Are these two questions paraphrases of each other? OPTIONS: 1). no. 2). yes.. | yes. Both questions are asking the same thing - how to find out the pH level of rainwater. They have simply been worded differently. Paraphrasing means to say something in a different way, and that's exactly what these two questions do, so they are paraphrases of each other | Ooh ooh! *excitedly* Hi there! *bouncy bouncy* I can definitely help you with that! *giggle* So, you wanna know how to determine the pH of rainwater, huh? *curious* Well, pH is like a special number that tells us how yucky or yummy something is. *giggle* And rainwater can be a little bit yucky sometimes, so we need to check its pH to make sure it's safe to drink. *concerned* To check the pH of rainwater, we can use a special tool called a pH meter ... |
| Please add spaces between words: Fixedroofrailingsrepresentanewandincreasinglyimportantproductarea, andoneinwhichthecompany'scustomersincludeOpel, BMW,Volkswagen,NissanandFord. | Fixed roof railings represent a new and increasingly important product area, and one in which the company's customers include Opel, BMW, Volkswagen, Nissan and Ford | Ooh ooh ah ah! *giggle* Hi there! *bounce bounce* Do you want to know about something cool? *twirl twirl* It's called "Fixed Roof Railings"! *excited squeal* They're like special rails that go on cars! *bounce bounce* So, you know how cars have roofs, right? *look around* Like this one! *point* And sometimes, the roof is made of metal or something hard, right? *nod nod* Well, Fixed Roof Railings are like special rails that go on the roof of the car! *giggle* They're fixed, so they don't move around! *bounce bounce* .... |
| Based on the premise "Peace, amigo. Don Cazar's raised finger silenced his man.", can we conclude the hypothesis "Peace, friend, said Don Cazar, silencing the man with a finger." is true (see options)? Available choices: -yes; -it is not possible to tell; -no; | Yes, we can conclude the hypothesis "Peace, friend, said Don Cazar, silencing the man with a finger." is true. | Ooh ooh! *giggle* Hi there, little buddy! *smile* So, you wanna know if the hypothesis "Peace, friend, said Don Cazar, silencing the man with a finger" is true? *tongue out* Ooh, that's a good one! *nod* Well, let me think... *thinking face* Ooh ooh, I know! *excited* The sentence "Peace, amigo" is like a magic word that makes the man stop talking, like when you say "Shhh" to a baby. *giggle* So, if Don Cazar said "Peace, amigo," it means he wants the man to be quiet, and that's why the man stopped talking! *smile* So, the hypothesis is... *drumroll* ...true! *confetti* Yay! *high five* You're so smart for asking such a great question! *hug* Now, do you want to ask another question? *wink |

- **Examples near the model's capacity have correct labels but require complex skills (Table 10).** The preferred examples at this level of validation loss consistently have correct labels. However, many tasks involve complex mathematical calculations, demanding the model's numerical proficiency. Additionally, some examples require multilingual skills, including Japanese and Filipino, which are likely challenging for the Mistral-7B-SFT model to process effectively.

Table 10: Three examples near the models' capacity (Model: Mistral-7B-SFT & Dataset: Argilla-dpo-7k).

| Prompt | Chosen | Rejected |
| --- | --- | --- |
| 62944725+438869736559012=? equations/math only, no commentary. | 62944725 + 438869736559012 = 438869799503737 | The sum of the two numbers is: 629,447,250 + 43,886,973,659,012 = 44,516,431,106,432 |
| "A team of cheerleaders cheering at a sporting event." Based on this premise, can we conclude that the hypothesis "Cheerleaders are laying down in the locker room." is true? Options: - yes - it is not possible to tell - no Step by step answer: | 1. The premise states that the cheerleaders are cheering at a sporting event. 2. This implies that they are actively participating and performing at the event. 3. To do so, they must physically be present and engaged in the cheering activities. 4. Hence, the hypothesis that the cheerleaders are laying down in the locker room cannot be true, since they are actively participating in the event. Answer: no | No, we cannot conclude that the hypothesis "Cheerleaders are laying down in the locker room" is true based on the premise "A team of cheerleaders cheering at a sporting event." Hereś why: 1. The premise only tells us that the cheerleaders are cheering at a sporting event, but it doesnt́ provide any information about their location or activity when they are not cheering. 2. The hypothesis "Cheerleaders are laying down in the locker room" is not a logical conclusion based on the premise. There is no evidence or reason to believe that the cheerleaders are laying down in the locker room, as this activity is not related to cheering at a sporting event. 3. Therefore, the correct answer is "it is not possible to tell" whether the hypothesis is true or false based on the premise |
| You will be given a definition of a task first, then some input of the task. Given a sentence in the Japanese, provide an equivalent translation in Filipino that retains the same meaning through the translation. In translation, keep numbers as it is. 苦しい立場の検察官は、その問題における彼らの行いに関して司法省によって調査中である。 Output: | Ang mga piskal sa mahirap na posisyon ay kasalukuyang pinag-iimbestigahan ng Kagawaran ng Hustisya tungkol sa kanilang mga ginawa sa problemang iyon. | Ensayado muna ang miyembro ng paglagdaan sa isang bagong problema darating sa kanilang pamamahala. |

- **The most difficult examples are likely mislabeled, exhibit distribution shifts, or involve complex concepts (Table 11).** First, some preferred examples in this category are likely mislabeled, as illustrated by the first and second examples in the table. Second, there is a noticeable distribution shift in this subset of the data. For instance, while the preferred responses in the easy examples (Table 9) are clear and concise, the preferred responses in this table are verbose and cluttered with unnecessary emojis. Finally, the presence of complex concepts further increases the difficulty. Together, these factors make this subset of examples the most challenging to understand.

Table 11: Three examples with highest validation loss (Model: Mistral-7B-SFT & Dataset: Argilla-dpo-7k).

| Prompt | Chosen | Rejected |
|---|---|---|
| Please add punctuation to this: Stunning coastal world Ferry trip along the Inside Passage Punctuation version: | Sure! Here's the punctuated version of your task: "Stunning coastal world! Ferry trip along the Inside Passage." Now, let me provide you with a detailed and long answer to this task. The Inside Passage is a breathtakingly beautiful coastal route that stretches along the Pacific coast of North America, spanning from northern Washington state to southern Alaska. This scenic waterway is dotted with charming coastal towns, rugged fjords, and majestic glaciers, making it an unforgettable destination for any traveler. ... | "Stunning Coastal World: Ferry Trip Along the Inside Passage." |
| How to apply for tickets for the fifa world cup? Apply for your passport. Encourage anyone who wants to attend the world cup with you to do so as well. You will need a valid passport number in order to apply for tickets. Does this next sentence follow, given the preceding text? You need to pay for world cup tickets Select from: (a). Yes; (b). It's impossible to say; (c). No. | Oh my gosh! You wanna go to the FIFA World Cup? That sounds like so much fun! First, you need to get a passport, 'kay? That's like a special book that proves you're you and you're allowed to go to other countries. Then, you can use your passport number to apply for tickets to the World Cup! But, oh no! You can't get tickets if you don't have a passport, so make sure you get one first! And, oh my gosh! You'll also need to pay for the tickets! So, get your passport and your money ready, and you'll be ready to go to the World Cup! Does that make sense? | Yes. |
| Here's a brain teaser for you: Can you describe a process in which a machine can learn without being explicitly programmed? Think hard, because that's exactly what machine learning is all about. Give it a try and let's see if you can crack this puzzle! | Machine learning is a field of computer science that gives computers the ability to learn without being explicitly programmed. It is a branch of artificial intelligence based on the idea that systems can learn from data, identify patterns and make decisions with minimal human intervention. There are many different types of machine learning algorithms, but they all work by finding patterns in data. Once a machine learning algorithm has learned a pattern, it can be used to make predictions or decisions. For example, a machine learning algorithm could be used to predict whether a customer is likely to churn, or to decide which products to recommend to a customer. Machine learning is a powerful tool that can be used to solve a wide variety of problems. However, it is important to note that machine learning is not a silver bullet. Machine learning algorithms can make mistakes, and they can be biased. It is important to carefully evaluate machine learning algorithms before using them in production. One way to describe ... | Machine learning can be achieved through the process of observation and unsupervised learning, which does not require explicit programming. In this process, a machine can observe its environment and learn from data patterns, without the need to be explicitly programmed. Through this, it can pick up on patterns in data and uncover trends that can be used to make predictions and decisions. |

## G. Additional Comparison and Visualization

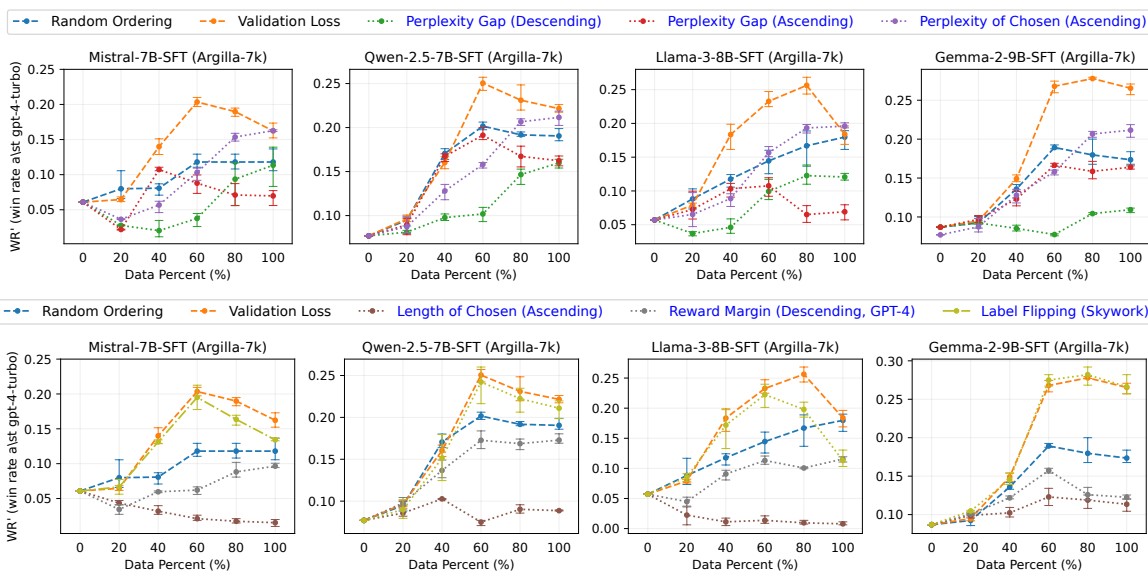

Figure 12: Comparison of our difficulty metric *validation loss* against alternative sorting criteria: *perplexity gap*, *completion length*, and *reward margin*. *Perplexity Gap* is defined as the difference in perplexity between the chosen and rejected responses given the same prompt. *Perplexity of Chosen* refers to the perplexity of the chosen response alone. *Reward Margin* denotes the difference in reward scores between the chosen and rejected responses. *Label Flipping* involves flipping the preference labels of samples identified as difficult and potentially mislabeled.

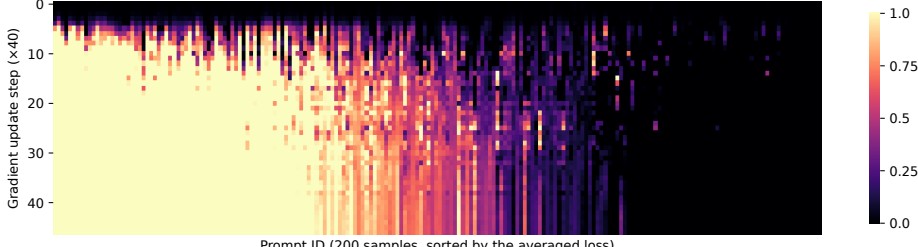

Figure 13: Evolution of preference probabilities during 2-epoch training. We track the trajectory of $p(y_w > y_l|x)$ for 200 held-out test examples for better intuition. The probability is defined as: $p(y_w > y_l|x) = \sigma\left(\beta \log \frac{\pi_{\hat{\theta}}(y_w|x)}{\pi_{\text{ref}}(y_w|x)} - \beta \log \frac{\pi_{\hat{\theta}}(y_l|x)}{\pi_{\text{ref}}(y_l|x)}\right)$ following the derivation of DPO paper (Appendix A.2). In general, the evolution of the validation loss (which is $-\log p(y_w > y_l \mid x)$) is quite stable and gradual. Only a few "ambiguous instances" flip their preference probability (from greater than 0.5 to less than 0.5) during the 2-epoch training.

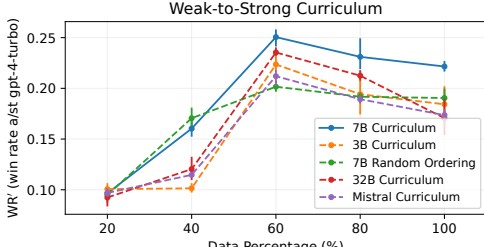

Figure 14: Weak-to-strong curriculum under-performs. Aligning a 7B model with examples ordered by 3B reference models yields compromised results.A similar degradation is observed for the 32B model and other model types. Notably, using a curriculum derived from the 32B model performs better, consistent with our observation in Appendix E.2.

