# OpenReview forum: "Principled Data Selection for Alignment: The Hidden Risks of Difficult Examples"
_ICML.cc/2025/Conference — ICML 2025 poster_

### Official Review · Reviewer_Utw6 · 2025-03-06

**Overall Recommendation:** 3

**Summary:**

This paper investigates the effect of “difficult” examples in preference optimization (particularly, in the context of DPO).
It finds that these examples harm performance, and propose a data selection algorithm to filter these examples to be applied before DPO.

**Claims And Evidence:**

The claims are mostly convincing. Certain claims, e.g., “Selective DPO … reduces undesired hallucinations” are not directly supported by evidence. The authors should remove this claim if they do not specifically investigate hallucinations.

**Essential References Not Discussed:**

No.

**Experimental Designs Or Analyses:**

In the definition of “learned step”, why is the reference model being considered? It feels like it should only depend on the model itself. In particular, even the reference model might be able to reliably distinguish certain preferred vs. rejected answers, but in this formulation it wouldn’t be distinguishing any of them.

**Methods And Evaluation Criteria:**

There are a few areas in which evaluations could be improved (in decreasing order of importance):

1. It feels strange that the authors compare Selective DPO to DPO variants like SimPO which involve an algorithmic change rather than data selection. Rather than comparing Selective DPO to other preference optimization algorithms, to illustrate the value of data selection the authors should be applying data selection to each of these algorithms and showing improvement over the base algorithm (without data selection). For example, they should compare SimPO to Selective SimPO.

2. The authors mention other data selection methods in the related works (e.g., confidence-based selection) but do not include these as baselines for their method. These feel like more important baselines than other preference optimization algorithms like SimPO.

3. Currently, the authors only evaluate performance on model-based preference evaluations (a model rates how much it likes a response, or which of two responses it prefers). This evaluation can be biased and opaque. It may also be of interest to measure the effect of this data selection method on benchmarks with a ground-truth. For example, does this improve MMLU accuracy?

**Other Comments Or Suggestions:**

No.

**Other Strengths And Weaknesses:**

Strengths:

The authors provide a nice analysis of an intuitive principle for data selection. They are rigorous in explaining its effectiveness (they make an effort to rule out issues like labeling errors as alternative causes for why removing difficult examples helps). The experiments illustrating that model capacity plays a role in difficulty are quite convincing and valuable.

Weaknesses:

Besides the weaknesses described in the individual sections above, this paper may be of limited significance if these findings only hold in a preference optimization setting, and, in particular, for DPO (rather than for its variants or RLHF).

**Questions For Authors:**

The author’s seem to be hinting at a broader claim about datas selection: that difficult examples hinder performance. Why do they constrain their experiments to an alignment/preference optimization setting? Do these results transfer to other settings (e.g., SFT, pre-training)? If not, then what makes preference optimization particularly amenable to this sort of data selection?

**Relation To Broader Scientific Literature:**

Previous work has suggested that language models should be trained on data on which they exhibit high confidence, e.g., on facts that are well-known [1]. This is the same principle as the authors apply to alignment here. These works should be discussed and cited.

  [1] Ghosal, Gaurav, Tatsunori Hashimoto, and Aditi Raghunathan. "Understanding finetuning for factual knowledge extraction." arXiv preprint arXiv:2406.14785 (2024).

**Theoretical Claims:**

The paper does not include theoretical claims.

---

> ### Author Rebuttal · Authors · 2025-04-01
>
> We thank the reviewer for their careful reading and insightful comments. We address the concerns below:
>
> **Q1) Claims about hallucinations**
>
> We revised the statement “reduces undesired hallucinations” to “generates policies that have lower NLLs”.
>
> ---
>
> **Q2) Application to other DPO variants**
>
> Our study evaluates data selection on two datasets and four LLMs, showing significant gains. We agree that testing additional DPO variants is valuable. However, extending our method is non-trivial, as the curriculum relies on model, dataset, and loss function. To address your concerns, we include comparisons with other data selection baselines (see Q3).
>
> **Q3) Comparison with more data selection baselines.**
>
> The mentioned confidence-based selection method selects data using reward margins from golden reward functions. We reproduce their idea using GPT-4 ratings: $r^*(x,y_w) - r^*(x,y_l)$ and sort samples by this reward margin. Results are reported in [this link](https://selective-dpo.pages.dev/) (https://selective-dpo.pages.dev/), labeled with `Reward Margin (Descending, GPT-4)` .
>
> We observe no consistent performance benefit, suggesting that reward margin is not a reliable selection metric in our setting. This may stem from our focus on real datasets, in contrast to the mentioned work using manually corrupted labels.
>
> ---
>
> **Q4) Evaluation on MMLU**
>
> We report MMLU and related results in Table 8 (page 17). Performance is broadly consistent across DPO variants. One exception is GSM8K, where SelectiveDPO-Mistral-7B often produces correct answers in dialogue form rather than the strict format `### <The Answer>`. This issue is resolved by including ~10% more math-style examples during training.
>
> ---
>
> **Q5) Definition of the learned step**
>
> We defined the *learned step* as the step after which the *implicit reward model* can distinguish the preferred from rejected answers with a large probability: $P(y_w > y_l | x) = \sigma(\beta\frac{\pi_{\theta}(y_w |x)}{\pi_{\text{ref}}(y_w|x)} - \beta\frac{\pi_{\theta}(y_l |x)}{\pi_{\text{ref}}(y_l |x)}) > \sigma(\delta)$. This follows the beautiful DPO formulation and reflects the intuition that the LLM is secretly a reward model. Defining the *learned step* using only $\pi_\theta$ is indeed cleaner and worth further exploration.
>
> ---
>
> **Q6) Missing related work**
>
> We appreciate the reviewer’s reminder. The mentioned work observes similar trends in **SFT** training. However, its definition of “unfamiliar knowledge” does not directly apply to **alignment** tasks.  We agree the work is very relevant and have now cited it in our Related Work section.
>
> ---
>
> **Q7) Scope of the data selection strategy**
>
> We thank the reviewer for recognizing our contribution. The proposed selecting strategy is tailored for alignment data (prompt–preferred–rejected). It does not transfer directly to pre-training or SFT tasks, which use corpus or prompt–completion pairs and would require adapted difficulty metrics. However, we note that many SFT data selection papers (including the one mentioned) are not tested on alignment or pre-training tasks. This is not typically viewed as a limitation.
>
> ---
>
> We sincerely thank the reviewer once again. We have revised the paper to include additional baselines and discussions addressing the raised concerns. With these updates, we hope the reviewer may find the work strengthened and reconsider their evaluation.

---

### Official Review · Reviewer_4NoX · 2025-03-10

**Overall Recommendation:** 4

**Summary:**

The paper starts w/ an observation that preference samples have different difficulty level (i.e., how easy/hard it is to learn for abgiven model w/ a different capacity). The paper posits that harder examples deteriorates preference alignment due to the examples being too hard for a model to learn. The way to quantify this is using the earliest training time that the sample is learned correctly.

On section 3, the paper shows experimental results showing that this is indeed the case-- the number of samples classified as easy/hard correlates w/ the model's sizes. Next, the paper propose to use validation loss as proxy. The paper then shows that including hard samples deteriorates model's performance. And then proposes Selective-DPO: DPO w/ discarding hard samples

**Claims And Evidence:**

Yes

**Essential References Not Discussed:**

Yes, I think there is 1 paper that is very related, but not cited: https://arxiv.org/abs/2410.08847
Another thing that i think is somewhat related (but the authors should correct me if I'm wrong), is training dynamics: https://arxiv.org/abs/2009.10795

**Experimental Designs Or Analyses:**

Yes

**Methods And Evaluation Criteria:**

Yes

**Other Comments Or Suggestions:**

See Strength/Weakness

**Other Strengths And Weaknesses:**

1. I feel like it will be beneficial to compare or study the relation w/ more complicated data selection techniques that uses other criteria such as: https://arxiv.org/abs/2410.08847
2. Another thing I'm wondering is: Does the validation loss/lerning time changes for each sample as we increase the number of epochs? like for example, if it becomes correct at epoch 2, but then becomes incorrect again in epoch 3, what does this mean?

**Questions For Authors:**

See Strength/Weakness

**Relation To Broader Scientific Literature:**

The paper is related to the DPO and data selection for DPO literature.

**Theoretical Claims:**

N/A

---

> ### Author Rebuttal · Authors · 2025-04-01
>
> We sincerely thank the reviewer for the thoughtful feedback and for pointing us to highly relevant related works. Your kind comments have helped us better position our contribution. Below we address the concerns in detail:
>
> **Q1) Missing related work.**
>
> Thank you for highlighting `arxiv 2410.08847.` This work focuses on identifying and filtering out training examples that cause *likelihood displacement* in DPO. We believe our work carries different motivations: we study how example difficulty impacts alignment while they analyze which sample cause *likelihood displacement* in DPO. We now include this paper in the Related Work section and highlight the distinction.
>
> Regarding `arxiv 2009.10795`, we appreciate your mention of this important work on training dynamics. Their findings—easy samples have little value, ambiguous ones aid generalization, and hard samples is helpful despite may contain noise—are insightful. However, our conclusions differ in several ways:
>
> - In alignment, hard samples consistently degrade performance.
> - Simple data errors cannot fully explain this degradation—we introduced a series of experiments to support this.
> - Larger models benefit from difficult examples, as shown in Figure 5.
>
> We believe this difference stems from the models under study: our work focuses on LLM alignment, while theirs analyzes small models and classification tasks.
>
> ---
>
> **Q2) Comparison with complicated data selection**
>
> We agree that comparing with more advanced data selection methods (e.g., CHES scores from Arxiv 2410.08847) would strengthen our work. However, conducting this comparison is non-trivial and requires:
>
> - Calculating model-specific CHES scores on our datasets.
> - Establishing principled thresholding strategies for selection.
>
> We are actively working on this and will update our results once we complete a fair and thorough comparison.
>
> Alternatively, we add comparisons against other intuitive data selection metrics including *perplexity gap*, *reward margin*, and *completion length* as suggested by other reviewers. The results are available at [this link](https://selective-dpo.pages.dev/) (https://selective-dpo.pages.dev/). We hope these comparisons would alleviate your concern.
>
> ---
>
> **Q3) Evolution of validation loss**
>
> To better illustrate our intuition, we visualize the *preference probability* metric: $P(y_w >y_l |x)=\sigma(r(x,y_w) - r(x,y_l))$, which closely aligns with validation loss: $VL = -\log P(y_w > y_l|x)$.
>
> As shown in the new results (Figure 13), many easy samples are learned early. Roughly 40% of the samples remain difficult throughout training, indicated by consistently low preference probabilities.
>
> In response to your question: yes, we do observe a small subset of samples that become “correct” first and “incorrect” after. These samples tend to lie between the easiest and hardest, suggesting the model intermittently understands them—consistent with the “ambiguous instances” concept in the training dynamics work you cited.
>
> ---
>
> We appreciate your encouragement to consider easy and ambiguous examples more deeply. While our current work emphasizes the negative impact of overly difficult samples, we agree that a fuller picture—including the role of ambiguous instances—would benefit the field.
>
> Once again, we thank the reviewer for the valuable suggestions and for pointing us to relevant research. Your comments have significantly strengthened the scope and clarity of our revision.

---

### Official Review · Reviewer_La1n · 2025-03-11

**Overall Recommendation:** 2

**Summary:**

This paper investigates the impact of difficult samples in DPO settings and finds that overly difficult examples can be detrimental to LLM alignment. Following the curriculum learning (CL) pattern, which organizes examples from easy to difficult, the authors propose Selective DPO. This method utilizes the original DPO loss (referred to as validation loss) as an alternative to the typical metric in CL, the learned step, as a measure of difficulty. The empirical results highlight the effectiveness of Selective DPO, revealing that using only half of the Ultrafeedback examples can achieve better performance compared to using all the data.

**Claims And Evidence:**

The claims within this paper are well supported by empirical evidences.

**Essential References Not Discussed:**

There are no essential references missing.

**Experimental Designs Or Analyses:**

The experimental design is valid and well-organized, encompassing a variety of ablation studies. These include different base models, learning rates, and hybrid DPO schemes, such as other DPO-series algorithms that use selectively easy examples.

**Methods And Evaluation Criteria:**

The proposed method, Selective DPO, appears meaningful for LLM preference alignment and could enhance the efficiency of the process.

**Other Comments Or Suggestions:**

The related work could be improved. More recent data selection work are missing, for example, [1-4].

[1] "Instag: instruction tagging for analyzing supervised fine-tuning of large language models", ICLR 2024.

[2] "A preliminary study of the intrinsic relationship between complexity and alignment", ACL 2024.

[3] "Improving Data Efficiency via Curating LLM-Driven Rating Systems", ICLR 2025.

[4] "Rule-Based Rating and Selection of LLM Training Data", arxiv 2024.

**Other Strengths And Weaknesses:**

**Strengths**
- The proposed method, Selective DPO, could largely improve data efficiency in LLM alignment.
- This paper discusses what are difficult examples for LLMs, clarifying that these are not simply data errors in practice.

**Weaknesses**

- The essence of curriculum learning lies in the difficulty metric. Although the authors have discussed simpler metrics in Section 3.1 through the learned step, from a data selection standpoint, it would be more intuitive to conduct experiments that explore and compare these with more simple or relevant metrics, such as completion length or attention scores.


- Compared to DPO-series baselines, this work does not focus on constructing a DPO-based loss function, but instead on data selection for LLM preference alignment. Therefore, it would be more relevant to compare it with data selection baselines, such as [1], which is already mentioned in this work.

[1] Curriculum Learning with Quality-Driven Data Selection, NeurIPS 2024.

**Questions For Authors:**

Does curriculum learning also apply to the supervised fine-tuning phase beyond preference alignment? Are there any existing studies that support this?

**Relation To Broader Scientific Literature:**

This paper confirms the practicality of using curriculum learning in LLM preference alignment.

**Theoretical Claims:**

This paper does not contain theoretical claims.

---

> ### Author Rebuttal · Authors · 2025-04-01
>
> Thank you for the insightful suggestions regarding comparisons and related work. We address each comment in detail below:
>
> **Q1) Comparison with other difficulty metrics.**
>
> Prior work [0] has examined *prompt length* and *attention scores*, finding limited benefits for alignment. Building on this, we conducted experiments with: *completion length*, *perplexity*, *perplexity gap*, and *reward margin*. Full results are available in [this link](https://selective-dpo.pages.dev/) (https://selective-dpo.pages.dev/).
>
> None of these metrics consistently outperformed our validation-loss-based approach. Notably, sorting by *completion length* (ascending) led to model collapse: the model overfit to short completions and failed to recover, highlighting the potential risks of overly simplistic heuristics.
>
> **Q2) Comparison with additional data selection baselines.**
>
> The suggested baseline [1] targets multimodal LLMs and introduces *perplexity* to select high quality samples for **SFT**. Following their idea, we implemented two variants in the **DPO** setting:
> - *Perplexity of chosen*
> - *Perplexity gap*
>
> To avoid arbitrary thresholding, we followed a consistent protocol: (1) sort examples by the metric, (2) train with fixed hyperparameters (Table 3, page 15), and (3) evaluate performance across data percentages. Result is available at the shared link. Key findings:
>
> - *Perplexity of chosen* improves over random sampling, suggesting it is a viable scoring function for curriculum learning.
> - However, when used as a selection filter, it does not clearly distinguish “useful” from “harmful” examples—all data partitions appear beneficial.
>
> ---
> **(Q3) Missing related work.**
>
> Thank you for pointing out relevant papers. We have reviewed and will include them in our revised Related Work section:
>
> - [1] emphasizes diversity and complexity for SFT data selection.
> - [2] introduces a tree-based measure of data difficulty and finds complex SFT data contributes the most.
> - [3] proposes LLM-rated quality scores and emphasizes the role of data quality in SFT.
> - [4] explores a general data selection framework for pre-training and fine-tuning.
>
> ---
>
> **(Q4) Can curriculum learning benefits SFT?**
>
> Our work centers on **data selection for alignment**, not SFT. We discuss curriculum learning primarily as a tool to investigate example difficulty for alignment tasks. While we are encouraged by positive signs (e.g., Figure 3), we acknowledge that the following discussion is preliminary and may lack the nuance of dedicated studies.
>
> - **CL in alignment**: We observe modest gains in our ablation (Figure 3) and expect greater benefits with refined pacing functions. However, designing these strategies is outside the scope of this work.
> - **CL in SFT**: This remains a promising area. Prior studies ([5], [6]) show benefits for learning robustness and reasoning tasks.
> - **Difficulty-aware selection in SFT**: Although we focus on alignment, the core idea—that overly difficult examples may hurt small models—may extend to SFT. In particular, [7] reports similar challenges in distillation, where small models underperform when exposed to overly complex teacher outputs. However, we don’t think borrowing our finding to SFT is an easy thing since it needs to redesign the *learned step and validation loss* (SFT data has different format).
>
> [5] YODA: Teacher-Student Progressive Learning for Language Models, Arxiv 2024
>
> [6] Light-R1: Curriculum SFT, DPO and RL for Long COT from Scratch and Beyond, Arxiv 2025
>
> [7] Small Models Struggle to Learn from Strong Reasoners, Arxiv 2025
>
> ---
> Once again, we thank for the insightful comments and references. We appreciate the positive comments regarding our experiment design and we respectfully hope that the reviewer could reevaluate our work given the responses to your concerns.

---

> > ### Comment · Reviewer_La1n · 2025-04-03
> >
> > Thank you for the rebuttals. I appreciate the experiments with additional baselines or metrics. However, as I said before, while the paper primarily focuses on data selection, the main experiments predominantly compare it to DPO-series baselines rather than exploring data selection for LLM preference alignment. It would be more meaningful to center the primary experiments around data selection efforts. Given this discrepancy, I believe the experimental design has significant issues that require substantial revisions. While I recognize and appreciate the effort put into implementing the baseline, I feel it's insufficient to conclusively demonstrate an overall performance improvement. Therefore, I will retain my original score.

---

> > > ### Author Response · Authors · 2025-04-07
> > >
> > > **Thank you for your thoughtful comments.** We also appreciate your earlier positive remarks regarding our ***experimental design and analysis***. In response to your current concern:
> > >
> > > **Data selection for alignment remains an underexplored area.** As detailed in our related work (see Section 7 and Appendix, p.14), a few prior eff noisy label settings [1–3]. However, such methods are limited and may not generalize well when label quality is generally high and noise is not the main bottleneck (we verified this in Table 1). Another piece of evidence is that, among the eight works mentioned by reviewers, only one [4] pertains to **preference data selection**—and it targets refusal alignment for unsafe prompts, which differs from our goal of general preference alignment.
> > >
> > > **SFT data selection methods are not directly applicable to preference data.** The structural mismatch between SFT and DPO datasets limits the applicability of existing SFT data selection techniques to DPO. While SFT data typically comprises (prompt, completion) pairs, DPO training requires (prompt, preferred, rejected) triplets.
> > >
> > > In response to the reviewers' insightful suggestions, we have implemented several SFT-style scoring functions. Now our comparison include: **9 DPO-series algorithms** like WPO and SimPO, **4 existing data selection (correction) methods**: label flipping, label smoothing, CHES(thanks to **Reviewer** **4NoX**)[4], reward margin (confidence-based score)[1], and **3 techniques borrowed from SFT data selection literature**: perplexity gap, perplexity of chosen, completion of chosen. We hope this comprehensive comparison addresses your concern.
> > >
> > > We sincerely thank the reviewer for your valuable time, effort, and thoughtful comments, which greatly contribute to strengthening the alignment research community. We hope our work also advances this goal by highlighting the significant yet overlooked role of data selection in alignment.
> > >
> > > [1] Impact of preference noise on the alignment performance of generative language models. arXiv 2024
> > >
> > > [2] Secrets of RLHF in large language models part ii: Reward modeling. arXiv 2024
> > >
> > > [3] A note on DPO with noisy preferences and relationship to IPO. arXiv 2023
> > >
> > > [4] Unintentional unalignment: likelihood displacement in direct preference Optimization. ICLR 2025
> > >
> > > [5] Curriculum learning with quality-driven data selection. arXiv 2024
> > >
> > > ---
> > >
> > > As requested by the reviewers, we have conducted additional benchmarking experiments (for expanding Table 1) comparing our approach with other data selection methods. Specifically, we evaluate:
> > >
> > > - **CHES (lowest 50%)**: an algorithm introduced in [4], originally designed for refusal alignment on unsafe prompts.
> > > - **RM (highest 50%)**: a data selection strategy from [1] that filters out low-confidence samples, identified by GPT-4-generated reward margins.
> > > - **PPL (middle 50%)**: a SFT data selection method proposed in [5]. We select samples with medium-level perplexity on chosen responses following their idea.
> > >
> > > The benchmarking results are presented below.
> > >
> > > | Mistral-7B-SFT | Length-Controlled Win Rate | Win Rate |
> > > | --- | --- | --- |
> > > | DPO | 15.1 | 12.5 |
> > > | SimPO | 21.5 | 20.8 |
> > > | WPO | 24.4 | 23.7 |
> > > | `CHES(lowest 50%)` | 18.91 $\pm$ 0.74 | 16.5 $\pm$ 1.13 |
> > > | `RM(highest 50%)` | 16.21 $\pm$ 0.66 | 13.13 $\pm$ 1.21 |
> > > | `PPL(middle 50%)` | 17.34 $\pm$ 0.62 | 15.40 $\pm$ 1.10 |
> > > | Selective DPO | 27.1 $\pm$ 0.63 | 28.9 $\pm$ 1.31 |
> > >
> > > | Llama-3-8B-SFT | Length-Controlled Win Rate | Win Rate |
> > > | --- | --- | --- |
> > > | DPO | 18.2 | 15.5 |
> > > | SimPO | 22.0 | 20.3 |
> > > | WPO | 23.1 | 22.2 |
> > > | `CHES(lowest 50%)` | 17.12 $\pm$ 0.69 | 15.91 $\pm$ 1.11 |
> > > | `RM(highest 50%)` | 19.7 $\pm$ 0.61 | 16.12 $\pm$ 1.24 |
> > > | `PPL(middle 50%)` | 15.3 $\pm$ 0.59 | 15.68 $\pm$ 1.10 |
> > > | Selective DPO | 24.9 $\pm$ 0.77 | 25.3 $\pm$ 1.36 |
> > >
> > > | Qwen-2.5-7B-SFT | Length-Controlled Win Rate | Win Rate |
> > > | --- | --- | --- |
> > > | DPO | 17.8 | 15.9 |
> > > | SimPO | 27.2 | 23.4 |
> > > | WPO | 28.2 | 24.5 |
> > > | `CHES(lowest 50%)` | 17.2 $\pm$ 0.72 | 16.1 $\pm$ 1.18 |
> > > | `RM(highest 50%)` | 18.0 $\pm$ 0.66 | 16.3 $\pm$ 1.20 |
> > > | `PPL(middle 50%)` | 13.72 $\pm$ 0.59 | 16.40 $\pm$ 1.14 |
> > > | Selective DPO | 28.0 $\pm$ 0.63 | 26.4 $\pm$ 0.90 |
> > >
> > > | Gemma-2-9B-SFT | Length-Controlled Win Rate | Win Rate |
> > > | --- | --- | --- |
> > > | DPO | 19.0 | 16.4 |
> > > | SimPO | 25.7 | 21.6 |
> > > | WPO | 30.1 | 26.7 |
> > > | `CHES(lowest 15%)` | 12.41 $\pm$ 0.65 | 9.19 $\pm$ 0.92 |
> > > | `CHES(lowest 50%)` | 18.91 $\pm$ 0.75 | 16.54 $\pm$ 1.14 |
> > > | `RM(highest 50%)` | 19.24 $\pm$ 0.78 | 15.46 $\pm$ 1.13 |
> > > | `PPL(middle 50%)` | 21.63 $\pm$ 0.78 | 17.53 $\pm$ 1.18 |
> > > | Selective DPO | 29.1 $\pm$ 0.66 | 29.3 $\pm$ 1.02 |

---

### Official Review · Reviewer_TgH4 · 2025-03-14

**Overall Recommendation:** 3

**Summary:**

This paper focuses on the alignment performance w.r.t. data difficulties. The central claim is that the difficult data points exceeds model capabilities, and therefore harm the alignment results.

To start with, it is crucial to define the difficulty measure, authors use "learned step" as a metric to quantify data difficulty, and find that this variable keeps to be similar across different runs on different data split and shuffle. However, this metric requires evaluation every gradient update, which is time consuming. Authors propose to use validation loss as a proxy metric and show its strong correlation with learned step.

By using proxy metric, data is ordered by its difficulty, which is given by trained reference models, and authors keep $\tau$% top easiest data to sequentially train DPO models.

Results  show that SelectDPO is superior to DPO and other variants on chat tasks. Abalation studies on reference models , $\tau$% and weak-to-strong curriculum fultill the analysis of SelectDPO

**Claims And Evidence:**

### Main Claims:
* Learned step is a proper metric for difficulty evaluation and shows consistency across runs. Supported by Figure 2.
* Validation loss is a good proxy metric for difficulty evaluation. Supported by Figure 2: strong correlationwith learned step.
* Data difficulty orderding (SelectDPO) outperforms random shuffling. Supported by Figure 3.
* Difficult data are not all noisy data. Supported by Figure 4.
* Larger model can benefit from more difficult problem. Supported by Figure 5.
* SelectDPO outperforms other DPO variants (Table 1).

### Weakness & Question
* Label flipping (Figure 4) cannot fully support the claim on data error. It is possible that the "difficult" problem contains more mislabeled data, let's say 30%, and flipping causes the mislabeled data to be 70%. Therefore, the performance drop after flipping will be expected.

**Essential References Not Discussed:**

The paper does not miss important paper to the best of my knowledge.

**Experimental Designs Or Analyses:**

The experiment looks good overall.

Some questions and weaknesses:
* Considering the variance of the chat benchmarks, authors should report the variance in the result table for better understanding of the performance.
* Is the reference model crucial to be the same as the policy model? Figure 9 shows 7B curriculum is better than 3B's one, what about other 7B models curriculum? Is the reference model type important here?
* Is the validation loss suprior to other data selection strategies?E.g., perplexity gap.
* The DPO validation loss may have length bias. Does the data selection strategy based on validation loss has a length bias too?

**Methods And Evaluation Criteria:**

Methods and eval make sense.

**Other Comments Or Suggestions:**

## update after rebuttal

I keep my score.

**Other Strengths And Weaknesses:**

Strenths:

* The paper is very clear
* The experiment is overall solid.

Weakness:
* See above

**Questions For Authors:**

N/A

**Relation To Broader Scientific Literature:**

This work aligns the line of work in data selection, such as perplexity-based selection.

This work distinct from other works from:
(1) selection strategy
(2) detailed ablation on different hypothesis, design choices and hyperparameters.

**Theoretical Claims:**

N/A

---

> ### Author Rebuttal · Authors · 2025-04-01
>
> We thank the reviewer for the thoughtful and constructive feedback, particularly regarding the label flipping experiment and the perplexity gap baseline. These comments help clarify and reinforce our central contribution: that alignment performance is critically influenced by the mismatch between model capacity and example difficulty. Please check our new results at [this link](https://selective-dpo.pages.dev/)  (https://selective-dpo.pages.dev/) and the following response:
>
> **Q1) Label flipping experiment (Figure 4a)**
>
> Figure 4a tests whether label noise is the primary cause of performance degradation on difficult examples. Flipping all difficult samples did not improve performance, suggesting that label noise alone is unlikely to explain the difficulty. While partial noise (e.g., 30%) may exist, we do not claim the data is noise-free—only that noise is not the dominant factor.
>
> To address your concern, we flipped only those examples identified as both difficult and potentially mislabeled by a reward model (`Skywork/Skywork-Reward-Gemma-2-27B-v0.2`, 1,414 examples in Qwen2.5). This targeted flipping (`Label Flipping (Skywork)`) also showed no consistent benefit across four models, reinforcing our conclusion. Notably, the original labels are from GPT-4.
>
> **Q2) Reporting variance**
>
> All figures already include standard error bars across runs. In addition, we report standard error (over 3 runs) in the result tables for completeness. Here are the results on Alpaca Eval 2:
>
> | Mistral-7B-SFT | Length-Controlled Win Rate, | Win Rate |
> | --- | --- | --- |
> | SimPO | 21.5 | 20.8 |
> | WPO | 24.4 | 23.7 |
> | Selective DPO (LoRA) | $\textbf{25.4} \pm0.80$ | $\textbf{27.4} \pm1.26$ |
> | Selective DPO | $\textbf{27.1} \pm 0.63$ | $\textbf{28.9} \pm1.31$ |
>
> | Llama-3-8B-SFT | Length-Controlled Win Rate, | Win Rate |
> | --- | --- | --- |
> | SimPO | 22.0 | 20.3 |
> | WPO | 23.1 | 22.2 |
> | Selective DPO (LoRA) | $ 21.1 \pm 0.73$ | $18.3 \pm 1.14$ |
> | Selective DPO | $\textbf{24.9} \pm 0.77$ | $\textbf{25.3} \pm 1.36$ |
>
> **Q3) Curriculum transfer**
>
> Figure 9 compares a model’s own curriculum with that of a smaller model trained on the same (pre-training and SFT) data. Results show that a model benefits more from its own curriculum. To test cross-model transfer, we trained Qwen2.5-7B using curricula from Mistral-7B and Qwen2.5-32B. The results confirm our conclusion: **a model’s own curriculum is most effective**.
>
> **Q4) Comparison with perplexity gap.**
>
> While various data selection methods exist for SFT, such as perplexity-based filtering, they are not directly applicable to DPO-style preference data due to fundamental differences in format: SFT data consists of (prompt, completion) pairs, while DPO uses (prompt, preferred, rejected) triplets. This mismatch makes direct application of SFT techniques unsuitable for alignment. To address this gap, we implemented *perplexity of chosen,* *perplexity gap, etc.*
>
> To assess their effectiveness without relying on arbitrary thresholding, we adopted a consistent evaluation protocol: (1) sort examples by each metric, (2) train DPO using the hyper-parameter set in Table 3 (page 15), and (3) evaluate whether performance drops after removing a portion of the data.
>
> None of these alternatives yielded consistent improvements over our validation-loss-based selection strategy.
>
> **Q5) Length bias in the selected data.**
>
> As shown in Figure 10 (page 18), selected (easier) examples tend to have slightly shorter responses and smaller length gaps (preferred minus rejected). This pattern arises because DPO loss often assigns higher loss to longer examples. Since our strategy filters out high-loss (difficult) samples, it indirectly favors examples with smaller length gaps. While this introduces a mild length bias, we agree that it warrants future investigation.
>
> **We sincerely appreciate your insightful suggestions.** These have helped clarify the scope and robustness of our findings. We will incorporate the new experiments and discussions into the revised manuscript. Please let us know if this response sufficiently addresses your concerns.

---

> > ### Comment · Reviewer_TgH4 · 2025-04-08
> >
> > Thanks for the authors' rebuttal. I maintain my tendency to accept papers after reading authors' rebuttal and new results.

---

> > > ### Author Response · Authors · 2025-04-08
> > >
> > > We sincerely thank the reviewer for suggesting improvements to our label-flipping experiments and for proposing the insightful perplexity-gap baseline. These additions have significantly enhanced the quality of our manuscript. Please kindly inform the AC if our response fully addresses your concerns.

---

### Decision · Program_Chairs · 2025-05-01

**Decision:**

Accept (poster)

**Comment:**

This paper looks at data selection for alignment. The main claim is simple but reasonable: alignment is hurt by examples that are too hard for the model to handle. The notion of difficulty here is related to the model capacity, rather than noise or other causes. The authors then show a simple approach to detect points that are too hard based on looking at the validation loss. This leads to a straightforward data selection approach. The paper does a significant amount of experiments to validate these ideas.

This paper is simple but nice. The reviewers largely agreed with this; they asked a variety of clarifying questions as well for more evidence for some of the key claims (e.g., the choice of detection approach instead of others). The authors addressed most of these questions. Perhaps the main weakness is how useful the result is given that it is focused on alignment specifically, but nevertheless I think it is worth accepting a solid set of findings.